EMBO
Molecular Medicine

# Targeted stabilization of Munc18-1 function via pharmacological chaperones

Debra Abramov[*] [ID], Noah Guy Lewis Guiberson [ID], Andrew Daab[†], Yoonmi Na, Gregory A Petsko[‡], Manu Sharma [ID] & Jacqueline Burré[**] [ID]

## Abstract

Heterozygous *de novo* mutations in the neuronal protein Munc18-1 cause syndromic neurological symptoms, including severe epilepsy, intellectual disability, developmental delay, ataxia, and tremor. No disease-modifying therapy exists to treat these disorders, and while chemical chaperones have been shown to alleviate neuronal dysfunction caused by mutations in Munc18-1, their required high concentrations and potential toxicity necessitate a Munc18-1-targeted therapy. Munc18-1 is essential for neurotransmitter release, and mutations in Munc18-1 have been shown to cause neuronal dysfunction via aggregation and co-aggregation of the wild-type protein, reducing functional Munc18-1 levels well below hemizygous levels. Here, we identify two pharmacological chaperones via structure-based drug design, that bind to wild-type and mutant Munc18-1, and revert Munc18-1 aggregation and neuronal dysfunction *in vitro* and *in vivo*, providing the first targeted treatment strategy for these severe pediatric encephalopathies.

**Keywords** Munc18-1; pharmacological chaperone; Rescue; small molecule; STXBP1

**Subject Categories** Neuroscience; Pharmacology & Drug Discovery

## Introduction

Heterozygous *de novo* mutations in the neuronal protein Munc18-1 (also known as STXBP1) were originally described in 2008 in five patients with Ohtahara syndrome, a severe infantile epileptic encephalopathy (Saitsu *et al*, 2008), and have since been linked to a series of neurodevelopmental disorders (Stamberger *et al*, 2016). Although many of these disorders include encephalopathies with epilepsy, such as Ohtahara and West syndromes (Saitsu *et al*, 2008; Otsuka *et al*, 2010), Dravet syndrome (Carvill *et al*, 2014), Lennox–Gastaut syndrome (Allen *et al*, 2013), and non-syndromic epilepsy

(Hamdan *et al*, 2009), mutations were subsequently found in disorders without epilepsy, such as ataxia–tremor–retardation syndrome (Gburek-Augustat *et al*, 2016) and intellectual disability without epilepsy (Hamdan *et al*, 2011; Stamberger *et al*, 2016). Overall, Munc18-1 encephalopathies are associated with epilepsy, severe to profound intellectual disability, developmental delay, ataxia, tremor, and other neurological symptoms (Saitsu *et al*, 2008; Milh *et al*, 2011; Stamberger *et al*, 2016; Suri *et al*, 2017). As individuals with neurodevelopmental disorders or intellectual disability without epilepsy are now being screened for Munc18-1 mutations, the number of cases is expected to significantly increase over the coming years (O'Brien *et al*, 2019). The only treatments that exist for Munc18-1-related syndromes are anti-epileptic drugs, which are unsuccessful in two-thirds of patients and do not lead to long-term improvements (Stamberger *et al*, 2016; Stamberger *et al*, 2017; Abramov *et al*, 2020). Additionally, anti-epileptics do not target the non-epilepsy symptoms, which exert a significant impact on the quality of life of patients and caregivers. Thus, development of a disease-modifying therapy is necessary to treat the diverse and wide-ranging symptoms of these syndromes.

Munc18-1 is a member of the Sec1/Munc18 family of proteins, which are required for secretory events throughout the cell (Toonen & Verhage, 2003). Munc18-1, in particular, is expressed in neurons and facilitates neurotransmitter release at the presynapse via its interaction with neuronal SNARE proteins (Rizo & Sudhof, 2012). Its knockout leads to complete abolition of neurotransmitter release, underscoring its importance for normal neurotransmission (Verhage *et al*, 2000). Despite extensive studies of Munc18-1's physiological role, it is not well understood how Munc18-1 dysfunction leads to the severe phenotypes seen in patients. Missense mutations of Munc18-1 result in destabilization and aggregation of the mutant protein (Saitsu *et al*, 2010; Martin *et al*, 2014; Chai *et al*, 2016; Guiberson *et al*, 2018; Kovacevic *et al*, 2018), which causes synaptic dysfunction *in vitro* and neuronal impairments in *C. elegans in vivo* (Guiberson *et al*, 2018). Similarly, haploinsufficiency which is predicted in patients with nonsense, splice-site, and frameshift mutations causes neuronal deficits (Orock *et al*, 2018; Chen *et al*, 2020). Importantly, we have recently shown that chemical chaperones rescue this dysfunction (Guiberson *et al*, 2018), albeit without

Appel Institute for Alzheimer's Disease Research, Brain and Mind Research Institute, Weill Cornell Medicine, New York, NY, USA
*Corresponding author. E-mail: dea2015@med.cornell.edu
**Corresponding author. E-mail: jab2058@med.cornell.edu
†Present address: University of Bath, Bath, UK
‡Present address: Ann Romney Center for Neurologic Diseases, Department of Neurology, Brigham and Women's Hospital and Harvard Medical School, Boston, MA, USA

Munc18-1 specificity and at high concentrations that may not be tolerable in humans. Thus, we sought to identify a targeted therapy to circumvent these drawbacks.

Here, we use an *in silico* screen followed by *in vitro* and *in vivo* experiments, to identify two pharmacological chaperones that bind and stabilize Munc18-1 protein levels both in mutant and hemizygous neurons. This molecular stabilization was accompanied by rescue of synaptic deficits and neuronal dysfunction in mouse neurons and in *C. elegans* models *in vivo*, providing the first targeted disease-altering strategy for these severe pediatric encephalopathies.

## Results

### Ligand-based rescue strategy

Disease-linked missense mutations in Munc18-1 result in reduced protein levels and increased aggregation, which in turn cause synaptic and neuronal dysfunction (Saitsu *et al*, 2010; Martin *et al*, 2014; Chai *et al*, 2016; Guiberson *et al*, 2018; Kovacevic *et al*, 2018). Thus, treatment strategies aimed at increasing functional protein levels are expected to alleviate these deficits. To test whether a ligand-based rescue strategy would work in general, we first tested the effect of the best-established ligand of Munc18-1, syntaxin-1, on Munc18-1 levels and solubility. Syntaxin-1 is a neuronal SNARE protein involved in synaptic neurotransmitter release that binds tightly to Munc18-1 (Hata *et al*, 1993; Pevsner *et al*, 1994). Syntaxin-1 contains a cytosolic domain with an N-terminal peptide (residues 1–28), followed by a conserved three-helix bundle designated as Habc domain (residues 29–180) and the H3 or SNARE domain (residues 180–264), which forms one of the four helices in the SNARE complex, and a C-terminal transmembrane domain (Fernandez *et al*, 1998; Rizo & Sudhof, 2012). Munc18-1 binds the N-terminal domain of syntaxin-1 via its domain 1, and the Habc and SNARE domain via its central arch (Fernandez *et al*, 1998; Burkhardt *et al*, 2008) (Fig 1A). Munc18-1 knockout mice express reduced syntaxin-1 protein, while other synaptic SNAREs are unaffected (Toonen *et al*, 2005), and knockout of syntaxin-1 causes a reduction in Munc18-1 levels (Vardar *et al*, 2016), suggesting that this interaction has specific impact on protein stability, and may work in the opposite direction as well.

We first tested the effect of syntaxin-1 on protein levels of four Munc18-1 mutations that we had previously studied: R406H, P480L, G544D, and G544V (Guiberson *et al*, 2018). We chose these mutants because multiple mutations at these sites cause disease (Stamberger *et al*, 2016), and they exhibit severe protein destabilization and aggregation (Saitsu *et al*, 2008; Martin *et al*, 2014; Chai *et al*, 2016; Hamada *et al*, 2017; Guiberson *et al*, 2018). Furthermore, analogous residues are mutated in the Munc18-1 paralog Munc18-2, which causes the immune disease familial hemophagocytic lymphohistiocytosis type 5 (Cote *et al*, 2009), potentially allowing for the extension of our study and rescue strategy to another disease. In HEK293T cells, which do not express endogenous Munc18-1 or syntaxin-1, we expressed either wild-type (WT) or disease-linked mutants of Munc18-1 in combination with two syntaxin fragments: Synt-1$^{1-180}$ or Synt-1$^{1-264}$ (Fig 1B). We found that co-expression of

Synt-1$^{1-264}$, which lacks only the transmembrane domain, increased total protein levels of all Munc18-1 mutants, as well as wild-type Munc18-1 (Fig 1C).

We then tested the effect of the two syntaxin-1 fragments on Munc18-1 solubility in the detergent Triton X-100 as a means to assess aggregation. We found both syntaxin-1 fragments to increase solubility of all mutant Munc18-1 variants tested (Fig 1D). Strikingly, the solubility of Synt-1$^{1-264}$ decreased in the presence of all four mutant Munc18-1 variants (Appendix Fig S1), highlighting the tight binding between these two proteins (Hata *et al*, 1993; Pevsner *et al*, 1994). Overall, these data establish that a rescue strategy for Munc18-1-linked encephalopathies based on ligand binding is feasible to reverse the structural deficits in disease-linked Munc18-1 variants.

### Structure-based drug design

We previously identified three chemical chaperones that were able to stabilize mutant Munc18-1 levels and solubility in primary neurons, as well as rescue neuronal deficits of mutant Munc18-1 in primary neurons and mutant UNC18 in *C. elegans* disease models (Guiberson *et al*, 2018). However, these chemical chaperones require a high dosage that may not be achievable in humans and do not bind specifically to Munc18-1, which may cause off-target effects (Cortez & Sim, 2014). Therefore, we performed an *in silico* structure-based screen to identify Munc18-1-targeted pharmacological chaperones.

The advantage of the *in silico* technique compared with other physical screening methods is that it not only enables screening of a vast library of compounds with varied chemical structures that may not be yet available, but also increases the hit rate of a follow-up physical screen from less than 0.01%, which is typical when a totally random library is screened, to 1–10% by biasing the experiment to compounds that are predicted to be likely to bind. The weakness of the *in silico* method is that the energy functions used to assess interactions are crude at best, do not take entropy into account, and cannot be used even to estimate the likely strength of binding. False positives may abound, and nothing may be known about false negatives; so overall, it is unknown what has been missed. Despite these caveats, structure-based drug design has been successfully used in multiple fields (Massa *et al*, 2006; Massa *et al*, 2010; Cazorla *et al*, 2011; Mecozzi *et al*, 2014; Gao *et al*, 2015; Lansu *et al*, 2017; Rivat *et al*, 2018; Li *et al*, 2020).

We identified three distinct sites in Munc18-1 that enable small molecule binding (Fig 2). Although small molecules that bind to site 3 may interfere with the ability of Munc18-1 to bind to syntaxin-1 (Fig 1A), we still screened for compounds that bind to this site, as the interference may be minimal depending on the affinity of the compound to Munc18-1. The binding sites for Munc18-1's other established binding partners, including Doc2 (Verhage *et al*, 1997), rab3 (Graham *et al*, 2008), and Mint1/2 (Okamoto & Sudhof, 1997), are not known.

We then used this optimized structure to perform a virtual screen of 255,780 compounds. The compounds were required to make three specific intermolecular contacts with Munc18-1 before being considered as hits. From the initial 255,780 compounds, seventeen of the compounds with the highest docking scores were commercially available and selected for *in vivo* screening (Appendix Table S2).

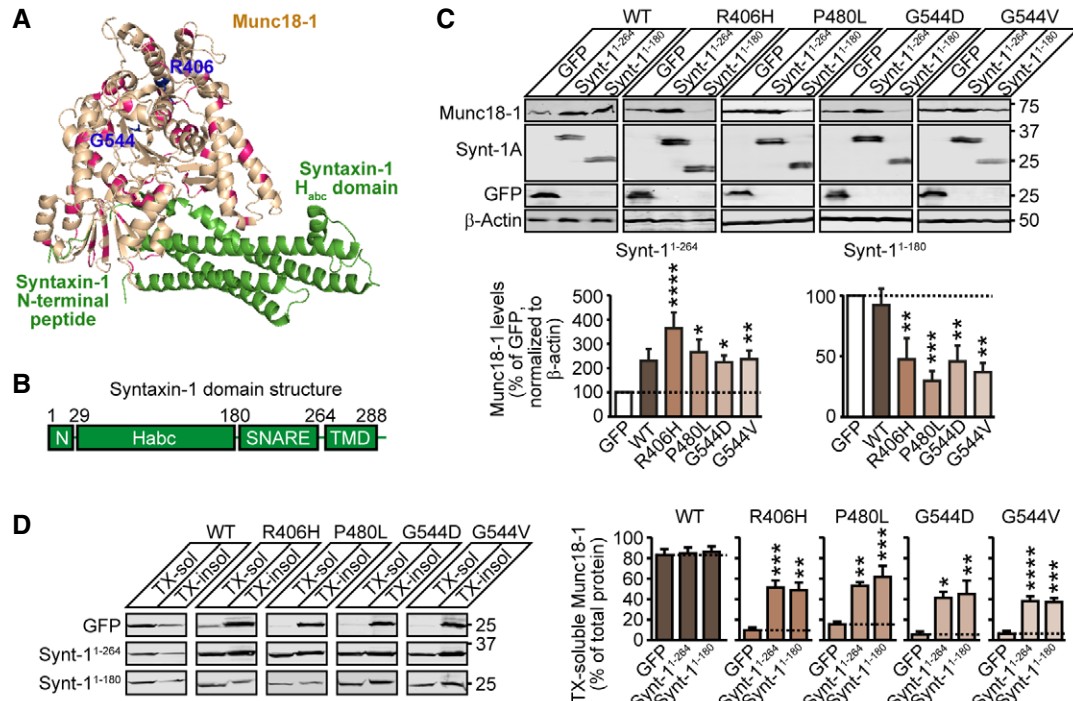

**Figure 1. Syntaxin-1 fragments rescue deficits in Munc18-1 levels and solubility.**

A   Localization of disease-causing missense mutations of Munc18-1b in its tertiary structure (PDB code 4JEU) (Burkhardt et al, 2008), with annotation of residues and binding sites of syntaxin-1 Habc domain and N terminus. Highlighted in blue are the residues 406 and 544 that we have analyzed in this study in detail.

B   Syntaxin-1 domain structure. The N terminus (residues 1-28) binds to domain 1 of Munc18-1 while the Habc domain (residues 28–180) and the SNARE domain (residues 180–264) bind in Munc18-1's central cleft. The transmembrane domain (TMD; residues 264–288) does not participate in Munc18-1 binding.

C   Total protein levels of Munc18-1. HEK293T cells transfected with WT or mutant Munc18-1 variants and either GFP, syntaxin-1$^{1-264}$ or syntaxin-1$^{1-180}$ were lysed and lysates were analyzed by quantitative immunoblotting to indicated proteins, normalized to β-actin (Synt-1A = syntaxin-1A). Data are means ± SEM (*$P < 0.05$, **$P < 0.01$, ***$P < 0.001$, ****$P < 0.0001$ by Kruskal–Wallis test, followed by Dunn's multiple comparison test; $n = 11$ independent experiments; exact $P$ values are shown in Appendix Table S1).

D   Solubility of Munc18-1. HEK293T cells transfected as in (C) were solubilized in 0.1% Triton X-100 (TX), and equal volumes of soluble and insoluble fractions were analyzed by quantitative immunoblotting. TX-soluble Munc18-1 was measured as percent of total Munc18-1 by quantitative immunoblotting. Data are means ± SEM (*$P < 0.05$, **$P < 0.01$, ***$P < 0.001$, ****$P < 0.0001$ by Kruskal–Wallis test followed by Dunn's multiple comparison test, or by one-way ANOVA followed by Bonferroni *post hoc* test; $n = 5$–9 independent experiments; exact $n$ and $P$ values are shown in Appendix Table S1).

Source data are available online for this figure.

Additionally, levetiracetam, a known anti-epileptic that is thought to bind to synaptic vesicle protein SV2a (Lynch *et al*, 2004), was included in our screen, as it provides a synaptic therapy that does not go through Munc18-1 and has led to seizure freedom in several patients with Munc18-1 mutations (Dilena *et al*, 2016; Stamberger *et al*, 2016; Liu *et al*, 2018).

**Total levels of Munc18-1 in primary neurons**

Disease-linked, heterozygous missense mutations in Munc18-1 result in reduced protein levels and cause reduction of wild-type Munc18-1 via a dominant negative mechanism, thereby reducing functional Munc18-1 levels well below 50% (Guiberson *et al*, 2018). Thus, stabilizing WT and/or mutant Munc18-1 may rescue identified deficits. We first tested how the compounds affect WT and mutant Munc18-1 protein levels in primary cortical neurons generated from conditional Munc18-1 knockout mice, in which exon 2 of the Munc18-1 gene is flanked by loxP sites and can be excised using cre recombinase (Heeroma *et al*, 2004). We infected

these neurons with lentiviral vectors expressing cre recombinase to drive knockout of Munc18-1. Simultaneously, we reintroduced R406H or G544D Munc18-1 via lentiviral expression. We chose to focus on R406H and G544D Munc18-1 as these mutants exhibited severe aggregation and impairment in neurotransmitter release in our previous work (Guiberson *et al*, 2018), multiple different mutations at residue 406 and 544 are linked to disease (Stamberger *et al*, 2016; Abramov *et al*, 2020), and different disease phenotypes are associated with these mutations (Stamberger *et al*, 2016; Abramov *et al*, 2020). Furthermore, these two mutations are located in domains distinct from each other and from the binding sites identified by our *in silico* screen. Additionally, we expressed cre recombinase in neurons derived from heterozygous conditional Munc18-1 knockout mice, to test compound effects on WT protein in hemizygous neurons.

At a concentration of 20 μM, only compound 13 significantly increased G544D Munc18-1 levels, similar to the chemical chaperone 4-phenylbutyrate (4-PBA) which we had previously shown to rescue Munc18-1 levels and neuronal deficits associated with

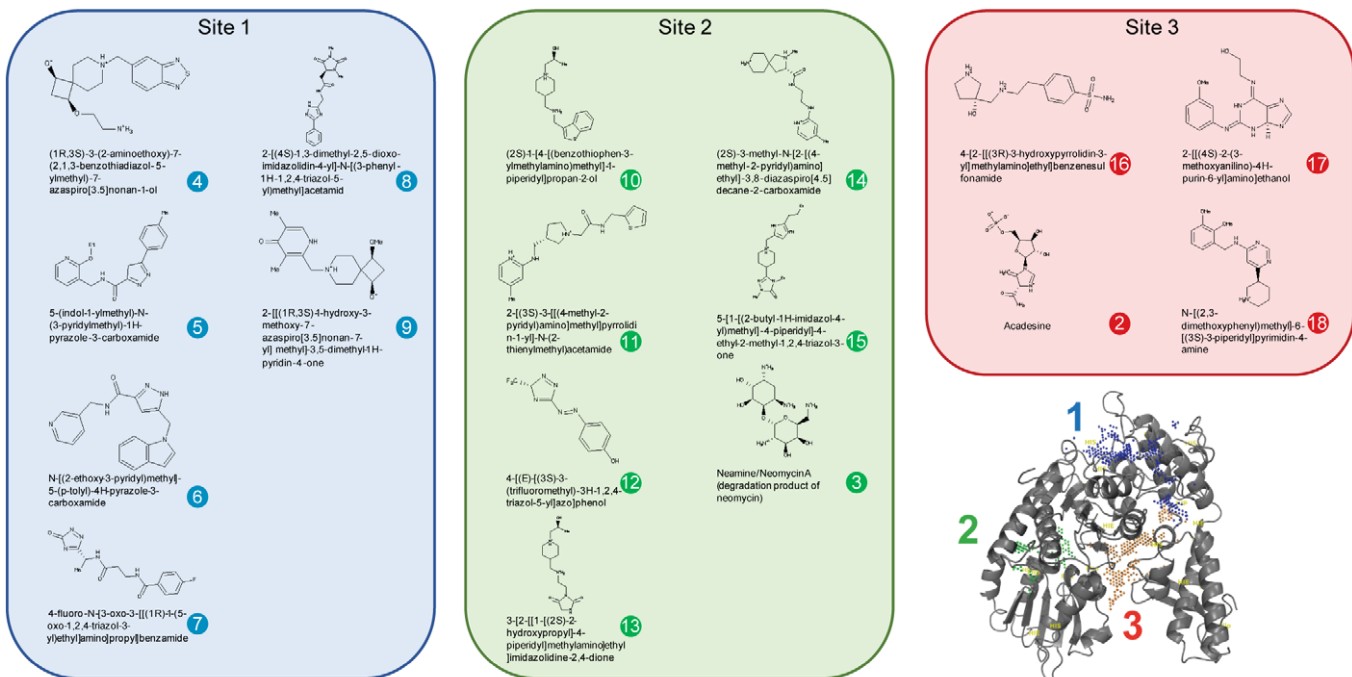

**Figure 2. Munc18-1-binding small molecules identified in the *in silico* screen.**

Localization of three binding sites is shown in the Munc18-1 tertiary structure (bottom right). Three potential binding sites, identified computationally by searching for pockets with favorable steric and electrostatic properties, were screened with the ZINC library. Chemical structures of top hits for each binding site are shown. Each compound was assigned a number for all following analyses (compound 1 = levetiracetam).

mutations in Munc18-1 (Guiberson *et al*, 2018) (Fig 3A and B and Appendix Fig S2A). In hemizygous Munc18-1 neurons, which serve as a model for mutations other than missense mutations, we found a significant increase for compound 9 (Appendix Fig S2B and C).

We then tested six select compounds at three additional concentrations on their effect on Munc18-1 G544D: We chose compounds 9 and 13 due to their advantageous effects at 20 µM, compound 10 as it has the same backbone structure as compound 13, compounds 14 and 16 because they demonstrated likely blood–brain barrier penetration as determined by an *in silico* algorithm (Liu *et al*, 2014), compound 11 due to the same scaffold structure as compounds 14 and 16, and because compound 16 had the overall best docking score. Compounds 9, 10, and 13 caused a significant increase in Munc18-1 G544D levels (Fig 3C and D), compounds 9 raised levels of Munc18-1 R406H (Appendix Fig S2D), and compounds 9 and 13 stabilized WT levels as well (Appendix Fig S2C).

**Direct binding of compounds to Munc18-1**

Are the observed rescue effects with compounds 9, 10, and 13 on mutant Munc18-1 levels due to direct binding to Munc18-1? To test this, we first purified recombinant GST, and GST-tagged WT, R406H, or G544D Munc18-1 (Appendix Fig S3) and measured conformational stability of WT and mutant Munc18-1 in the presence or absence of compounds using limited proteolysis under native conditions, where increased resistance to trypsin digestion represents increased structural stability. We found all compounds to increase WT, R406H, and G544D stability (Fig 4A and B),

mirroring our data in primary neurons (Fig 3 and Appendix Fig S2).

We separately measured direct binding of compounds to immobilized Munc18-1 variants where we incubated bead-immobilized GST alone, GST-tagged wild-type, R406H, or G544D with our compounds and measured bead-bound compound fluorescence. We found that all three compounds directly bind to WT and mutant Munc18-1 (Fig 4C). When we modeled binding of the three compounds to Munc18-1, we found interestingly that compounds 10 and 13 are predicted to bind with the same backbone in site 2, with the head group facing outwards (Fig 4D). Having two compounds with such similar chemical backbone not only provides additional confidence in our findings, but provides space to modify this compound to eventually facilitate its use as a novel therapeutic, i.e., for improvements in crossing the blood–brain barrier, or for tagging so the compound can be traced in real time to assess metabolic turnover and pharmacokinetics.

**Spontaneous neurotransmitter release**

Does the molecular rescue of Munc18-1 translate into a functional rescue? To test this, we assessed the ability of compounds 9, 10, and 13 to rescue synaptic deficits in neurons expressing mutant R406H or G544D Munc18-1, using a microelectrode array (Fig 5). We measured spontaneous neuronal activity in Munc18-1 knockout neurons expressing R406H or G544D Munc18-1 before and after addition of compounds. We found compounds 9 and 13 to

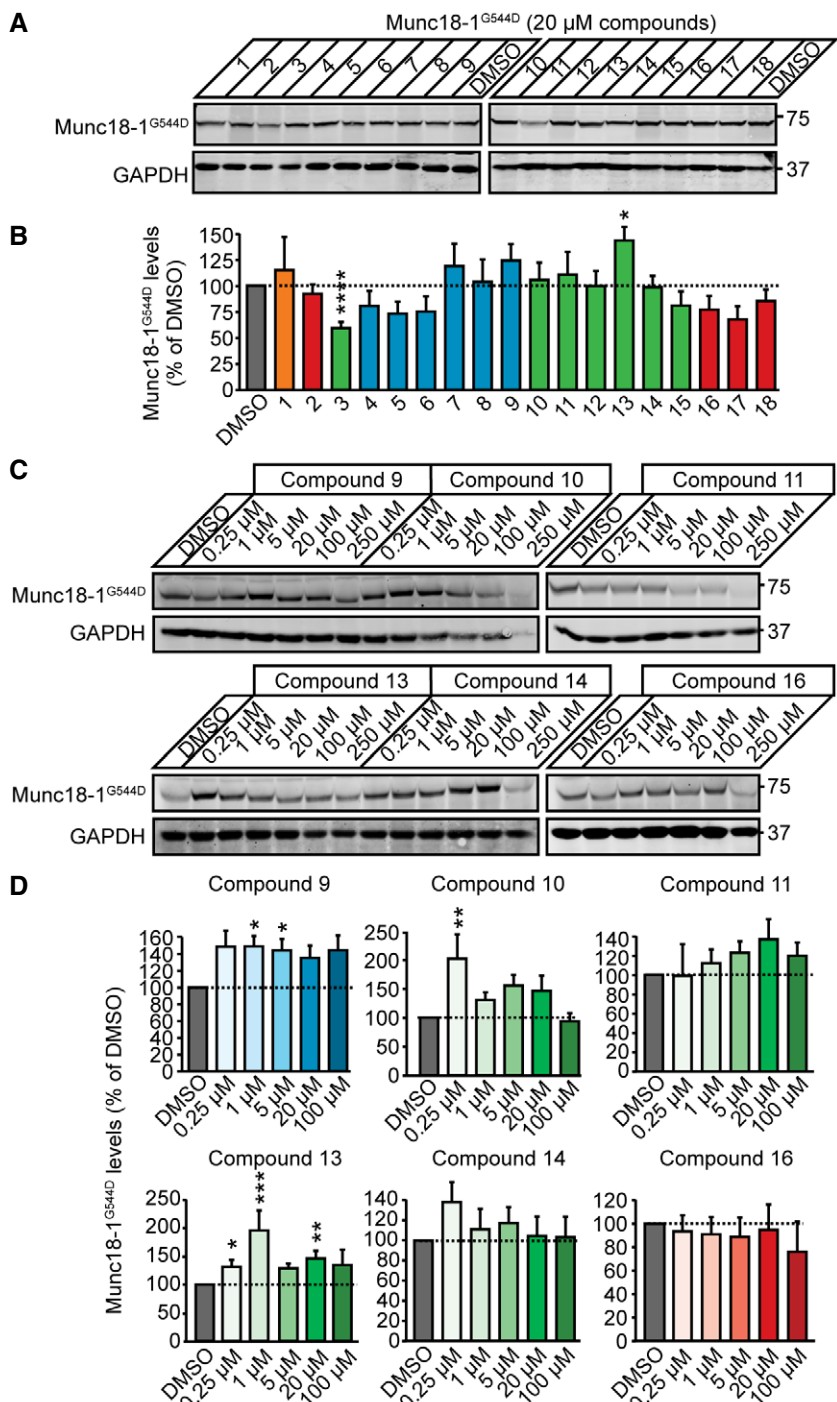

**Figure 3. Rescue of total levels of Munc18-1 G544D in primary neurons.**

A, B  Total protein levels of G544D Munc18-1. G544D Munc18-1b was expressed in cortical mouse neurons infected with lentiviral vectors expressing cre recombinase in the presence or absence of compounds at 20 μM, or DMSO (vehicle control). Total Munc18-1 levels were analyzed by quantitative immunoblotting 9 days after infections (A), normalized to GAPDH levels (B). Data are means ± SEM (*$P < 0.05$, ****$P < 0.0001$ by one-way ANOVA followed by Bonferroni *post hoc* test; $n = 6$ independent experiments; exact $P$ values are shown in Appendix Table S1).

C, D  Same as in (A, B) except that compounds 9, 10, 11, 13, 14, and 16 were added at 0.25, 1, 5, 20, 100, or 250 μM. Note, that the data for 250 μM were not plotted in (D) because of neuronal death. Data are means ± SEM (*$P < 0.05$, **$P < 0.01$ by Kruskal–Wallis test with Dunn's multiple comparison test, or one-way ANOVA with Dunnett's *post hoc* test; $n = 6$–15 independent experiments; exact $n$ and $P$ values are shown in Appendix Table S1).

Source data are available online for this figure.

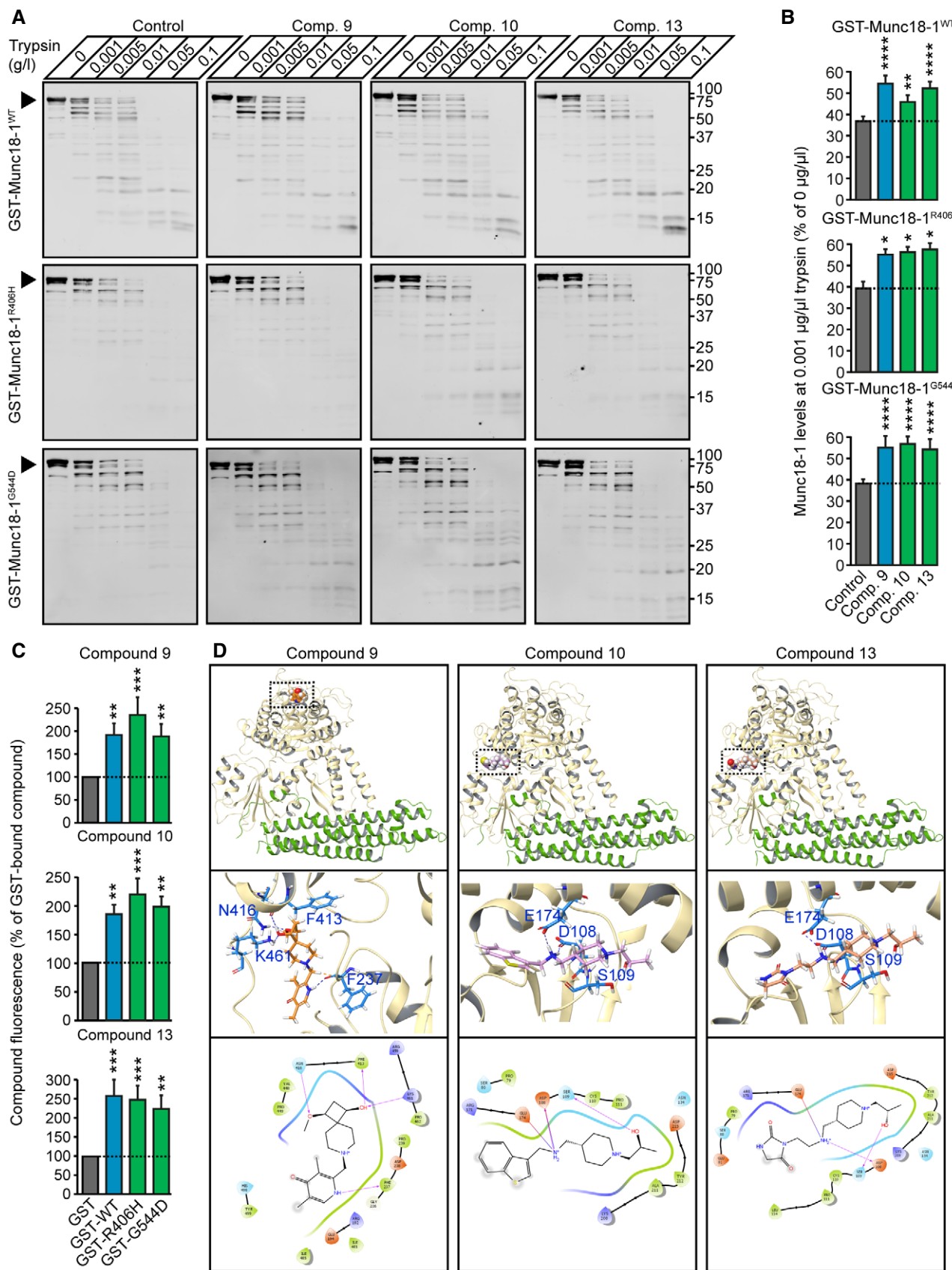

**Figure 4.**

**Figure 4.  Direct binding of compounds to recombinant Munc18-1 and modeling of binding mode.**

A, B   Limited proteolysis. Recombinant purified proteins were incubated with increasing concentrations of trypsin in presence or absence of 20 μM compound. Remaining protein levels were analyzed by quantitative immunoblotting. Data are means ± SEM (*$P < 0.05$, **$P < 0.01$, ****$P < 0.0001$, by two-way ANOVA and Dunnett's multiple comparison test; $n = 4$ independent experiments; exact $P$ values are shown in Appendix Table S1).

C   Direct binding of compounds to Munc18-1. Bead-bound fluorescence of compounds 9, 10, and 13 was quantified upon incubation of compounds with bead-immobilized GST, or GST-tagged WT, R406H or G544D. Data are means ± SEM (**$P < 0.01$, ***$P < 0.001$, by Kruskal–Wallis test and Dunn's multiple comparison test, or by one-way ANOVA followed by Dunnett's *post hoc* test; $n = 9$ independent experiments; exact $P$ values are shown in Appendix Table S1).

D   Predicted binding sites of compounds 9, 10 and 13. The top set of images gives a global view of the binding sites. The middle set of images gives a 3D view of residue intermolecular interactions with the compounds of interest. The bottom set of images gives a diagrammatic 2D representation of the same interactions. Hydrogen bonds are represented by arrows (arrowhead pointing to acceptor) and salt bridges by a red to blue gradient line (color change red:negative, and blue:positive).

Source data are available online for this figure.

significantly increase mean firing frequency in G544D Munc18-1 neurons, and compound 13 in R406H Munc18-1 neurons (Fig 5A and B and Appendix Fig S4). In contrast, compound 10 led to a significant reduction in mean firing frequency at high concentrations, mimicking its effect on total Munc18-1 levels in primary neurons (Fig 3 and Appendix Fig S2).

**Evoked synaptic vesicle cycling and neurotransmitter release**

To test the effect of our compounds on synaptic vesicle cycling, we used a synaptotagmin antibody uptake assay (Kraszewski *et al*, 1995). We have previously shown that Munc18-1 mutants dramatically reduce vesicle cycling as compared to WT Munc18-1

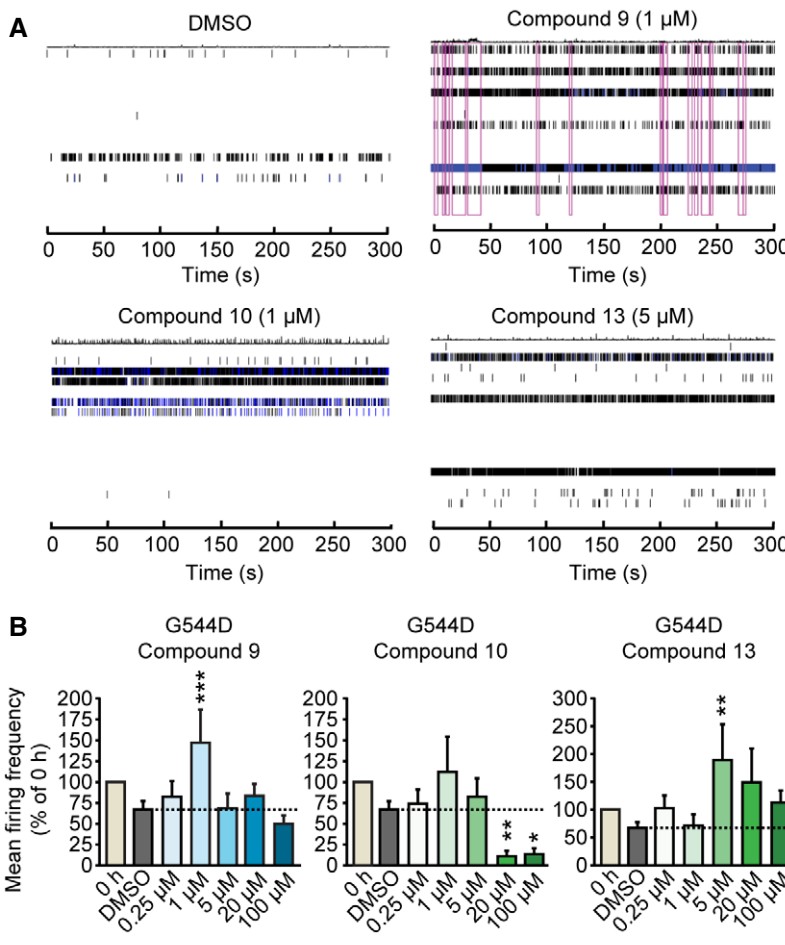

**Figure 5.  Rescue of spontaneous neurotransmitter release in cortical mouse neurons expressing Munc18-1 G544D.**

A, B   Munc18-1 knockout neurons expressing G544D Munc18-1 were plated on a microelectrode array and subjected to analysis of mean firing frequency before addition of compounds (0 h) or 48 h after vehicle (DMSO) or compound addition. 16 electrodes per well were analyzed for neuronal firing (A; purple boxes indicate network activity). Data are means ± SEM (*$P < 0.05$, **$P < 0.01$, ***$P < 0.001$ by one-way ANOVA and Dunnett's multiple comparison test, or Kruskal–Wallis test followed by Dunn's *post hoc* test; $n = 11$–16 independent experiments; exact $n$ and $P$ values are shown in Appendix Table S1).

Source data are available online for this figure.

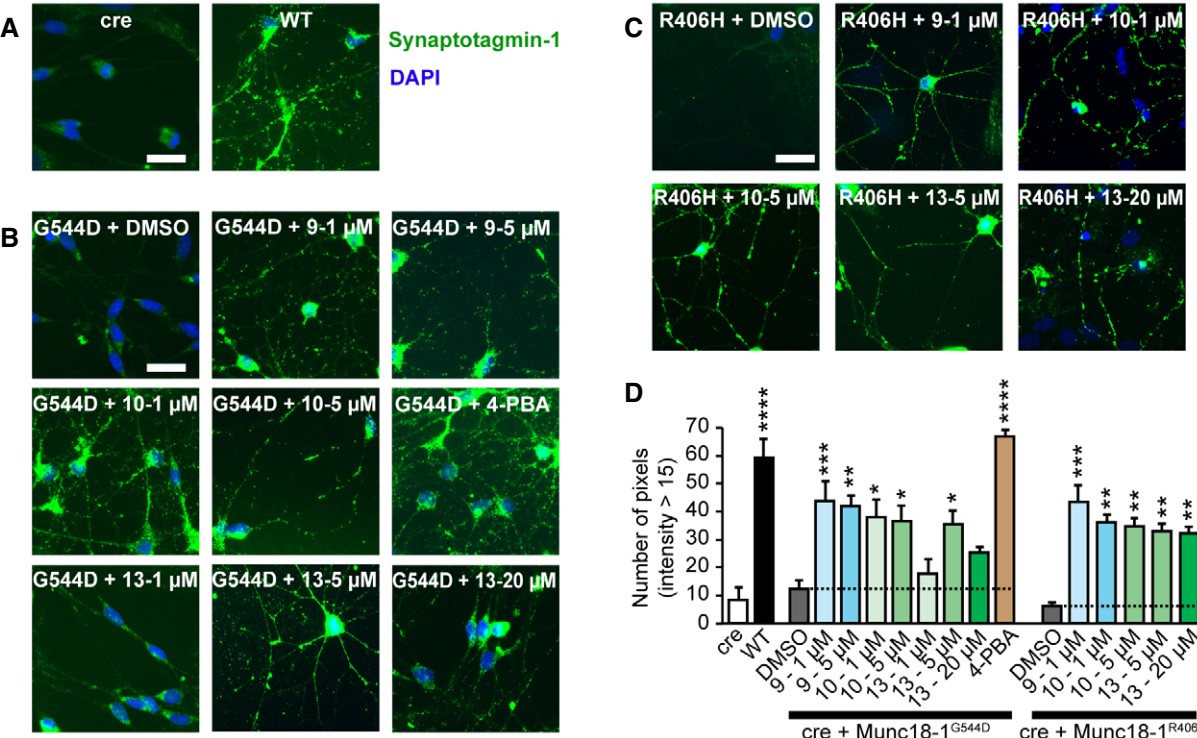

**Figure 6. Rescue of synaptic vesicle cycling in mouse cortical neurons expressing mutant Munc18-1.**

A–D  Uptake of synaptotagmin-1 antibody during high K$^+$ stimulation. Neurons expressing cre recombinase and/or WT, R406H, or G544D Munc18-1b with or without compound were subjected to an antibody uptake assay 7 days after lentiviral infection. Endocytosed synaptotagmin-1 antibody was quantified by immunostaining (A-C; scale bar = 30 μm; 4-PBA = 4-phenylbutyrate), via counting the number of pixels > intensity of 15 (D). Data are means ± SEM (*$P < 0.05$, **$P < 0.01$, ***$P < 0.001$, ****$P < 0.0001$ by one-way ANOVA and Dunnett's multiple comparison test; $n = 3$–10 independent experiments; exact $n$ and $P$ values are shown in Appendix Table S1).

Source data are available online for this figure.

(Guiberson *et al*, 2018). We first stimulated primary Munc18-1 null neurons (cre), WT neurons, or those expressing R406H or G544D Munc18-1 with high potassium and quantified the fluorescence intensity of endocytosed synaptotagmin-1 luminal domain antibody (Fig 6A–C). Neurons expressing only cre, R406H, or G544D Munc18-1 with DMSO vehicle control demonstrated low antibody uptake compared with WT neurons (Fig 6D and Appendix Fig S5). When we treated neurons with compounds 9, 10, and 13, we found a significant rescue of antibody uptake in R406H and G544D Munc18-1 neurons for all compounds tested (Fig 6 and Appendix Fig S5).

In addition to the antibody uptake assay, we measured evoked neurotransmitter release using changes in fluorescence of the synaptic vesicle-localized chimeric protein synaptophysin-pHluorin (Granseth *et al*, 2006). pHluorin is quenched at the acidic pH of synaptic vesicles and reveals an increase in fluorescence upon exposure to the neutral extracellular pH (Miesenbock *et al*, 1998; Sankaranarayanan & Ryan, 2000). When we subjected WT neurons to elevated potassium, we detected a sudden increase in fluorescence and a slow decline, representing synaptic vesicle exocytosis followed by endocytosis (Fig 7A). This activity was almost completely abolished in neurons expressing R406H or G544D Munc18-1 (Fig 7A–H). Addition of our three compounds at concentrations that were effective to boost Munc18-1 protein levels rescued the amount of exocytosis (Fig 7H and I). Overall, our data suggest that compounds 9 and 13 are effective at not only

stabilizing functional Munc18-1 levels, but also supporting Munc18-1's function in synaptic vesicle exocytosis.

## Rescue of neuronal function in live worms

We next assessed the effect of compounds 9, 10, and 13 on neuron function *in vivo*, in transgenic *C. elegans* strains expressing WT, R405H, or G544D UNC-18, the worm versions of human mutant Munc18-1, that we had previously generated (Guiberson *et al*, 2018). We had previously found faster paralysis of mutant worms under heat shock compared with WT worms, demonstrating the increased tendency of mutant UNC18 to misfold (Guiberson *et al*, 2018). We fed worms compounds 9, 10, or 13 at increasing concentrations for three generations and then quantified changes in their heat shock response. We found compounds 9 and 13 to significantly delay the heat shock paralysis of the R405H and G544D UNC-18 worms (Fig 8A–L and Appendix Fig S6), with compound 10 having little to no effect (Fig 8D and J).

Worms expressing GFP-tagged UNC-18 G544D reveal lack of axonal and dendritic localization of mutant UNC-18 and accumulation of somatic aggregates in the ventral nerve cord (Fig 8M), a phenotype that could partially be rescued by the chemical chaperone 4-phenylbutyrate (Guiberson *et al*, 2018) (Fig 8N and P). When we assessed the effect of compounds 9, 10, and 13 on the

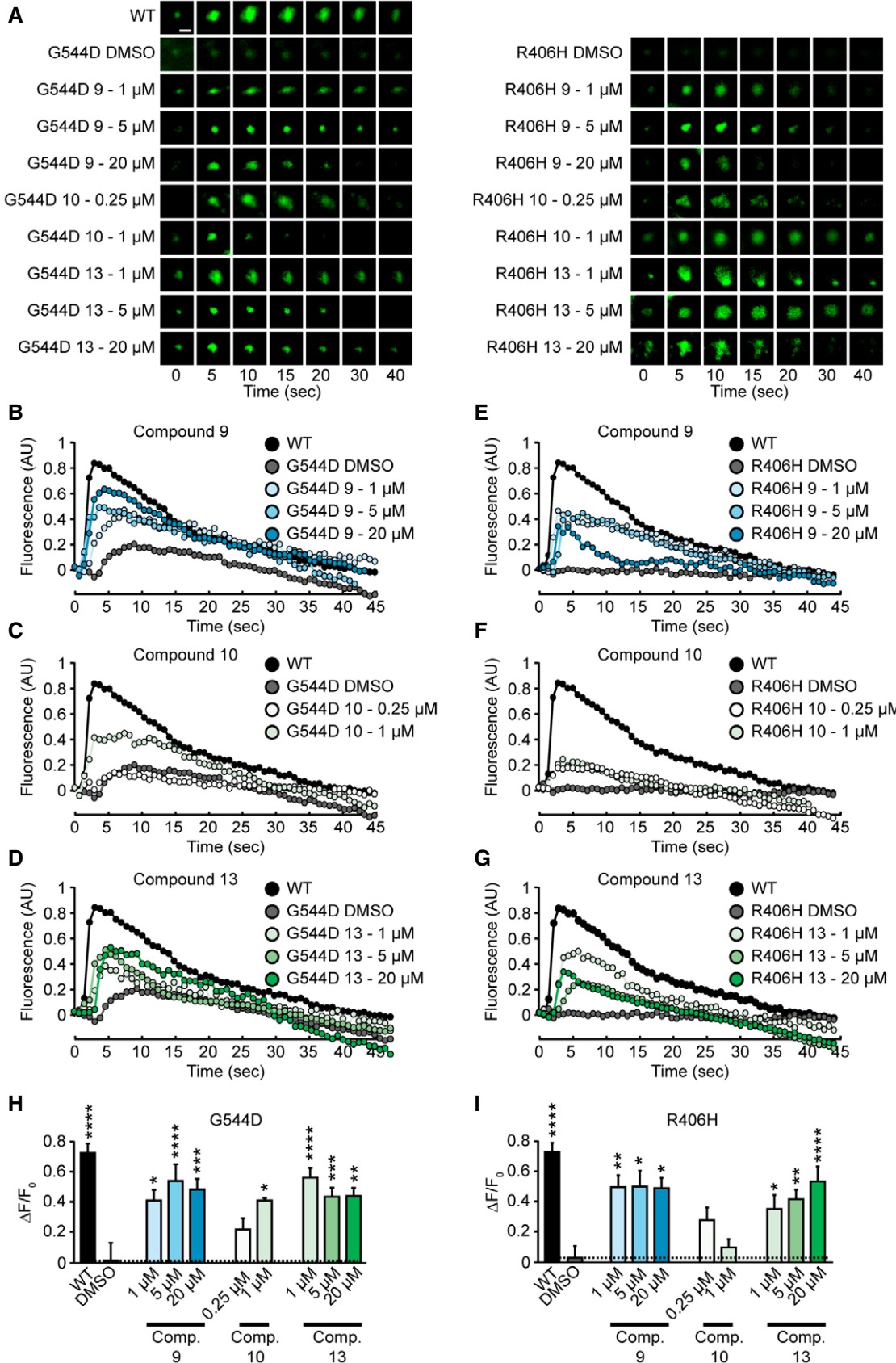

Figure 7.

**Figure 7.  Rescue of evoked neurotransmitter release in mouse cortical neurons expressing mutant Munc18-1.**

A–G   Representative images (A) or traces (B-D & E-G) of changes in fluorescence of synaptophysin-pHluorin upon stimulation of neurons (scale bar in panel (A) = 2 µm).

H, I    Plot of change in fluorescence from baseline to peak fluorescence for Munc18-1 G544D (H) or R406H (I). Data are means ± SEM (*$P < 0.05$, **$P < 0.01$, ***$P < 0.001$, ****$P < 0.0001$ by one-way ANOVA and Dunnett's multiple comparison test; $n = 7–23$; exact $n$ and $P$ values are shown in Appendix Table S1). Note that compound 10 is fluorescent at a similar wavelength as pHluorin, so experiments at concentrations higher than 1 µM were not possible.

Source data are available online for this figure.

subcellular localization of mutant UNC-18 in live worms, we found reduced aggregate formation and restoration of the cytosolic distribution of G544D UNC-18 at all compounds (Fig 8N–P).

### Connection of the rescue of Munc18-1 levels with functional rescue

To better highlight the effect sizes of compounds 9, 10, and 13, we compared the effects of the compounds on G544D and R406H Munc18-1 stability and function (Fig 9A and B). We determined protein stability as the average of each compound's effect on total protein levels in mouse neurons and protein aggregation in *C. elegans*, measured via protein thermolability in the heat shock paralysis assay. We compared this with the rescue of Munc18-1 function, measured as the averages of the compounds' effects on both spontaneous and evoked neurotransmission.

The plot highlights that for both mutants, compounds 9 and 13 work at 1 µM, 5 µM, and 20 µM concentration, whereas compound 10 had only minor effects at 1 and 5 µM. For both mutants, high concentrations of all compounds negatively affect Munc18-1 function, despite an increase in protein levels. We speculate that this is due to off-target effects of the compounds that may impair overall neuronal function.

The effects of compounds 9 and 13 on both mutants differ slightly: compound 9 is more potent in stabilizing R406H Munc18-1 (Fig 9B), whereas compound 13 shows higher efficacy on G544D Munc18-1 (Fig 9A). We speculate that this may be due to the domain where the mutation is located and where the compounds bind (Fig 1A and Fig 2). R406H and G544D are located in different domains, likely leading to differences in folding and/or misfolding. Stabilization of the area that is primarily destabilized is expected to have higher efficiency in stabilizing the entire protein, i.e., compound 9 for R406H and compound 13 for G544D. Overall, however, we see a positive correlation between increase in mutant Munc18-1 stability and Munc18-1 function.

## Discussion

Using structure-based drug discovery based on *in silico* docking experiments, we have identified two pharmacological chaperones that restore deficits in mutant Munc18-1 folding and levels, and that rescue neuronal and synaptic deficits in primary neurons and live *C. elegans*. To our knowledge, this is one of the few studies that directly translates from computational screening to having a reliable and robust effect both *in vitro* and *in vivo*. Several studies have successfully identified small molecule mimetics for BDNF from *in silico* studies that are able to either specifically activate or antagonize TrkB, the BDNF receptor, and are also active *in vivo* (Massa *et al*, 2010; Cazorla *et al*, 2011). Other studies have identified

pharmacological chaperones for the p75 receptor (Massa *et al*, 2006), neuronal FLT3 receptor (Rivat *et al*, 2018), Mas-related G protein-coupled receptor X2 (Lansu *et al*, 2017), and cyclin-dependent kinase 4/6 (Gao *et al*, 2015). However, identification of small molecules against pathogenic proteins that are not enzymes, channels, or receptors, has proven far more difficult, as these proteins do not have easily identifiable and functional binding sites. A recent study identified a small molecule that stabilizes the amyloid precursor protein (APP)-trafficking retromer protein complex in primary cortical neurons *in vitro* (Mecozzi *et al*, 2014), and more recently, in mice as well (Li *et al*, 2020), increasing retromer levels and reducing the accumulation of amyloid-beta and APP processing. Importantly, our study has identified two such compounds that are active *in vivo* and are able to stabilize both the WT and mutant Munc18-1 protein. The binding sites of the compounds are structurally distinct from the studied R406H and G544D mutations, thus likely translating to other missense mutations in Munc18-1. In addition, the stabilization of WT Munc18-1 by compounds 9 and 13 is expected to alleviate synaptic deficits in patients that are heterozygous for Munc18-1 due to nonsense, frameshift, and splice site mutations (Chen *et al*, 2020).

The clinical heterogeneity seen in patients with Munc18-1 mutations has made treatment very difficult. Although all patients exhibit intellectual disability and many have epilepsy, patients differ as to the types and severity of these seizures and other neurodevelopmental symptoms. Furthermore, seizure duration, severity, and age of onset are not associated with developmental outcome in patients with Munc18-1 mutations, meaning that therapies aimed at seizure control do not address the developmental aspects of the disease (Stamberger *et al*, 2017). Given that the prevalence of Munc18-1 variants among individuals with unspecified developmental disorders is 0.25–0.5% and the prevalence of intellectual disability in the general population is 1%, the number of individuals diagnosed with Munc18-1-associated neurodevelopmental disorders will likely rise substantially (O'Brien *et al*, 2019). Anti-epileptic drugs, which are currently the only type of treatment for patients with Munc18-1 mutations, therefore often fall short in the majority of patients. Thus, mechanistically targeted treatments specific to Munc18-1 are needed. Although chemical chaperones are theoretically a viable strategy for treating these diseases, they require concentrations that will likely have off-target effects due to their lack of specificity (Guiberson *et al*, 2018). The compounds identified here are specific toward Munc18-1 and have shown efficacy both in mouse neurons and in our *in vivo C. elegans* models at micromolar concentrations without lead optimization, which is promising and comparable to other small molecules identified via *in silico* screens (Mecozzi *et al*, 2014; Rivat *et al*, 2018). Future studies will focus on derivatization, to further enhance efficacy and improve pharmacokinetics across the blood–brain barrier, as well as on testing of these compounds in a hemizygous mouse model of disease, which mimics human

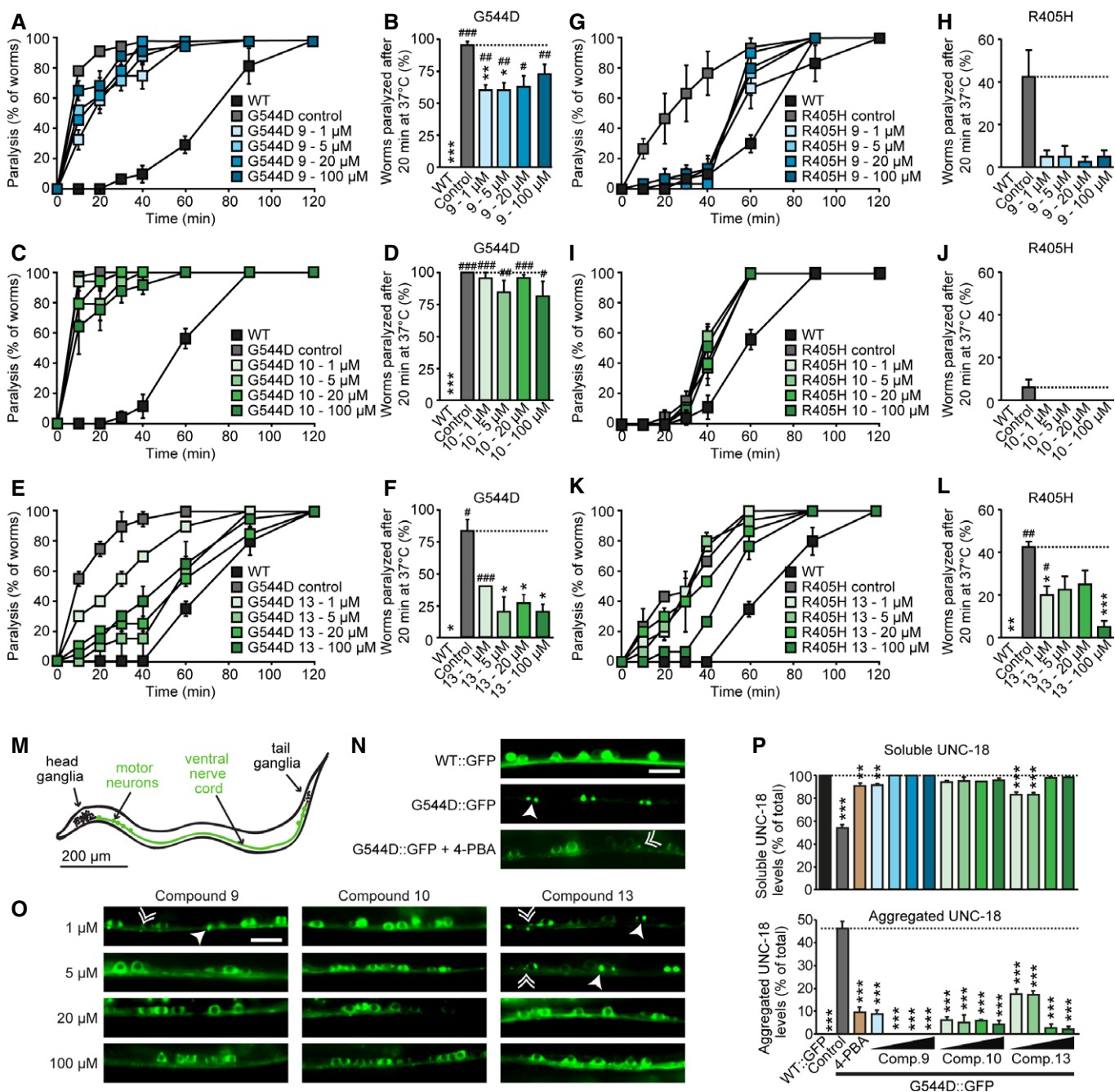

**Figure 8. Rescue of deficits in mutant *C. elegans*.**

A–L (A, C, E, G, I, K) Heat-induced paralysis. Indicated worm strains maintained in compounds at indicated concentrations were exposed to 37 °C over a period of 120 min, and paralysis was scored at indicated time points. Data are means ± SEM (*n* = 3–4 independent experiments on ten worms per experiment; exact *n* and *P* values are shown in Appendix Table S1). (B, D, F, H, J, L) Quantification of paralysis at 20 min. Data are means ± SEM (*,#*P* < 0.05, **,##*P* < 0.01, ***,###*P* < 0.001, as compared to WT, * as compared to mutant by two-way ANOVA and Dunnett's multiple comparison test; *n* = 3–4 independent experiments on ten worms per experiment; exact *n* and *P* values are shown in Appendix Table S1).

M Image of a worm, highlighting head and tail ganglia, as well as the ventral nerve cord and motor neurons.

N, O Rescue of the subcellular localization of UNC-18 in worms expressing G544D UNC-18. *C. elegans* expressing WT::GFP or G544D::GFP at 1, 5, 20, or 100 μM compound were immobilized, and the ventral nerve cord was imaged. Solid arrowheads point to pairs of bigger puncta, broken arrowheads to single, smaller puncta (N, O). Scale bar in (N) and (O) = 10 μm. 4-PBA = 4-phenylbutyrate.

P Quantification of soluble and aggregated UNC-18. Data are means ± SEM (***P* < 0.01, ****P* < 0.001 by one-way ANOVA and Dunnett's multiple comparison test; *n* = 3–4 worms; exact *n* and *P* values are shown in Appendix Table S1).

Source data are available online for this figure.

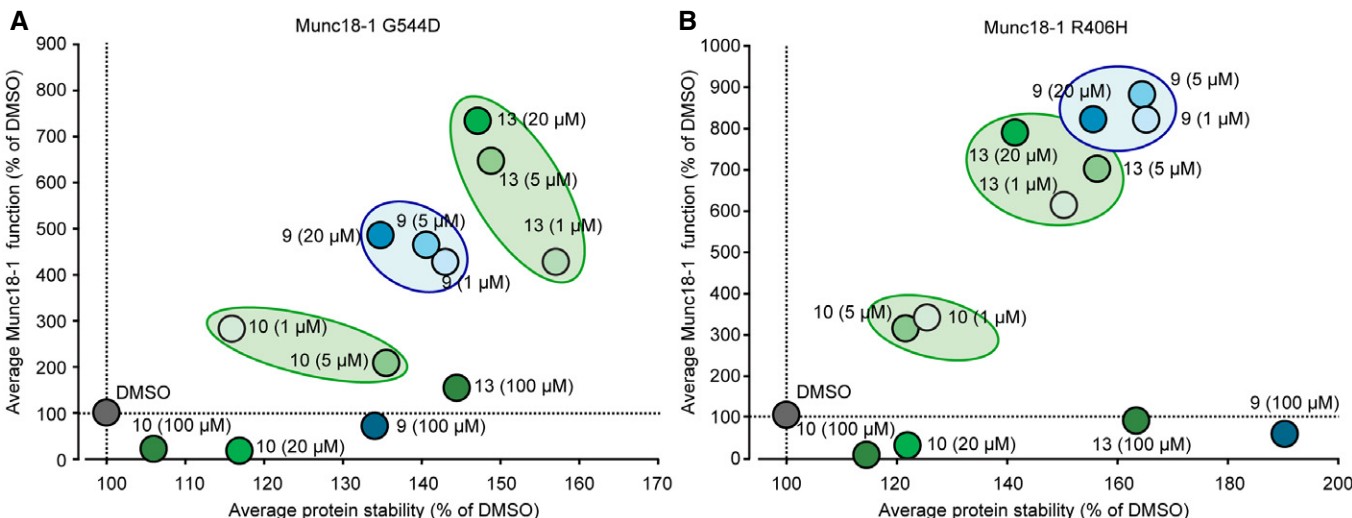

**Figure 9. Plot of the effects of compounds 9, 10, and 13 on G544D and R406H Munc18-1 function and stability.**

A, B Averaged protein stability was calculated from the data for total mutant protein levels and mutant *C. elegans* heat shock paralysis (Appendix Fig S2, 3 and 8). Averaged mutant Munc18-1 function was obtained from the data for MEA plate assay, antibody uptake, and synaptopHluorin release (Appendix Fig S4 and Figs 5–7). See Materials and Methods for details.

Source data are available online for this figure.

nonsense, frameshift, and deletion mutations and has a phenotype that overlaps with symptoms of Munc18-1 encephalopathies (Chen *et al*, 2020). For children with missense mutations in Munc18-1, testing of our compounds is required in a heterozygous mutant mouse model, which still needs to be developed. It is promising, though, that two of our compounds bind the same site with the same basic structure, while the variant part of these compounds faces toward the cytosol and could therefore be modified to alter stability and toxicity, blood–brain barrier penetration, or to label the compound with tags so they can be tracked. The superior rescue effect of compound 13 over compound 10 already gives clues on the pathway for derivatization.

We used three different assays to determine the effect of compounds on Munc18-1 function at the synapse. Comparison of the microelectrode array assay (spontaneous neurotransmitter release), with the uptake and pHluorin-based assays (evoked neurotransmitter release) indicates that the compounds rescue these types of release differently. In the evoked neurotransmission experiments, the quantifiable fluorescence signal comes from specific synapses that release neurotransmitter in mutant Munc18-1 neurons, meaning synapses that do not release neurotransmitter do not show up. In the microelectrode array assay, the mean firing frequency is an average of spontaneous somatodendritic currents (including those that do not result in downstream action potentials) of all neurons—those which spontaneously fire and those which do not during the course of the experiment. Overall, these assays measure different forms of neuronal activity, i.e., presynaptic release versus somatodendritic local field potentials, and the differences of results seen in these experiments are thus likely due to this technical difference.

Furthermore, it is puzzling that compound 10 rescued only evoked neurotransmitter release at low concentrations, but was unable to rescue spontaneous release. This discrepancy may be

due to the fact that a different synaptic machinery is involved in evoked versus spontaneous release (Kavalali, 2015). In particular, a known binding partner of Munc18-1, Doc2b, has been implicated in regulating spontaneous release (Groffen *et al*, 2010; Pang *et al*, 2011). Compound 10 may alter Munc18-1's association with Doc2b in a way that may affect spontaneous transmission, perhaps by affecting Doc2b's capacity to bind SNARE complexes. Alternatively, Munc18-1 has a large number of synaptic binding partners that may also be affected by binding of compound 10 to Munc18-1, and some of these may also be differentially involved in spontaneous versus evoked release. This effect of compound 10, despite its shared binding site with compound 13 and its similar chemical structure, may assist in further rounds of derivatization as a way of testing how compounds affect Munc18-1's various roles at the synapse.

Most interestingly, multiple diseases are linked to mutations in SM proteins. While Munc18-1 is associated with infantile encephalopathy with epilepsy and other neurodevelopmental disorders (Stamberger *et al*, 2016; Abramov *et al*, 2020), Munc18-2 is expressed in lymphocytes and other cells of the hematopoietic system, as well as epithelial cells, and mutations cause familial hemophagocytic lymphohistiocytosis type 5 (Cote *et al*, 2009). Munc18-3 is ubiquitously expressed, but reduced levels of Munc18-3 are linked to obesity (Garrido-Sanchez *et al*, 2013) and glucose intolerance (Oh *et al*, 2005; Bergman *et al*, 2008), mutations in VPS45 lead to severe congenital neutropenia (Vilboux *et al*, 2013), and mutations in VPS33B cause arthrogryposis–renal dysfunction–cholestasis (ARC) syndrome (Gissen *et al*, 2004). Despite differences in primary sequences, the overall 3D structure of these SM proteins is strikingly similar (Archbold *et al*, 2014), potentially enabling translation of our findings and usage of these three compounds also in other diseases.

# Materials and Methods

## *In silico* analysis

### Model preparation

The models of Munc18-1 for *in silico* screening were obtained from the PDB 3C98 (Burkhardt *et al*, 2008). These models were further optimized for screening using the Protein Preparation Wizard of the Maestro Software Suite (v. 9.9 Schrödinger LLC). The protein models were optimized to a reduced state in a pH of 7.4 and then analyzed for possible binding sites utilizing the Sitemap function of Maestro. The function produced a series of sites that could be used for *in silico* screening against any *in silico* chemical library.

### Grid generation

As part of the screening process, a series of coordinates was generated over the pre-prepared protein model. It is a grid over which to position the various compounds of interest. The grid was generated using the grid generation tool of Maestro (v. 9.9 Schrödinger LLC). No restraints were placed on the model before the grid was generated. The grids were generated to encompass all of the site maps generated previously.

### Compound library preparation

A series of compounds was obtained from the ZINC database. The compounds were then screened by Steven Trueman for redundancy, using molecular fingerprinting, in the chemical structures of the library. The resulting library consisted of 255,780 compounds. Using the Ligprep tool (version 2.9 Schrödinger LLC) in Maestro, these compounds were then given the proper ionization state at pH 7.4, and the rotatable bonds were optimized for minimal free energy in solution. The minimization was done using the OPLS_2005 force field. The new set of conformers was used for screening against the Munc18 proteins of interest.

### In silico screening

The pre-prepared protein models and the previously prepared *in silico* library were then used for *in silico* screening. The screening was performed using a series of methods in the Glide program (v 6.2 Schrödinger LLC) as part of the Maestro suite (Friesner *et al*, 2004). The first of the algorithms was the HTVS method. The default parameters were used to screen the set of compounds. Briefly, this involves keeping the protein model rigid and the compound models flexible. The compounds were required to make three specific intermolecular contacts with the protein model before being considered as hits. The scoring of each of the compounds correlates with the strength of the interaction of the compound for the protein. It is useful to eliminate early very weak candidates with the HTVS algorithm. That was done here by choosing to continue screening 500 of the compounds with the top docking score. Here, the top docking score is the one with the greatest negative docking score. These 500 compounds were screened for a second time with the standard precision (SP) method. In Maestro, the HTVS and SP methods share the same algorithm for screening. However, SP takes into account all its parameters. Screening with the SP method gave a list of compounds with another set of docking scores (Appendix Table S2).

## Mouse strains

The conditional Munc18-1 knockout mouse was kindly donated by Dr. Matthijs Verhage at CNCR (the Netherlands) (Heeroma *et al*, 2004). Mice were bred as homozygous conditional Munc18-1 knock-out mice on a CD-1 background, and P0 pups were used for generation of neuron cultures. No inclusion criteria for neuronal cultures were used, and mice were randomly assigned to virus and treatment groups. Mice were housed with a 12-h light/dark cycle in a temperature-controlled room with free access to water and food. All animal procedures were performed according to NIH guidelines and approved by the Committee on Animal Care at Weill Cornell Medicine.

## *C. elegans* maintenance and assays

Strains were generated by standard methods (Guiberson *et al*, 2018).

## *C. elegans* imaging

Animals were immobilized using a 30 mg/ml solution of 2,3-butanedione monoxime in M9 buffer (22 mM $KH_2PO_4$, 42.3 mM $Na_2HPO_4$, 85.6 mM NaCl, and 1 mM $MgSO_4$) and were then mounted on 2% agarose pads, and the ventral nerve cord was imaged on an Eclipse 80i upright fluorescence microscope (Nikon).

## *C. elegans* heat shock paralysis

After three generations of growth with or without compound at 1, 5, 20, or 100 μM, ten young adult animals were placed on agar plates and subjected to 37 °C in an incubator. All animals were confirmed to be capable of locomotion prior to the assay by observing their movement in response to a head-poke stimulus. At each time point, the percentage of paralyzed animals was assessed by their ability to move in response to a head-poke stimulus. Each genotype was coded, tested ~ 10 times blindly, and the paralysis curves were generated by averaging paralysis time courses for each plate.

## Rescue experiments

Mouse cortical neurons were incubated for 48 h with compounds at the indicated concentrations before analysis. Worms were kept for 2–3 generations on agar plates with OP50 bacteria containing compounds at the indicated concentrations. Worms were then either live imaged for aggregation studies or were subjected to heat shock paralysis.

## Cell culture and maintenance

HEK293T cells (ATCC) were maintained in DMEM with 1% penicillin/streptomycin and 10% bovine serum. Cells were solely used as protein expression systems or to produce lentivirus and have not been authenticated or tested for mycoplasma contamination. Mouse cortical neurons were cultured from newborn mice of either sex. No inclusion criteria for neuronal cultures were used, and mice were randomly assigned to virus and treatment groups. Cortices were dissected in ice-cold HBSS, dissociated and triturated with a siliconized pipette, and plated onto 24-well plastic dishes. Plating media (MEM supplemented with 5 g/l glucose, 0.2 g/l $NaHCO_3$,

0.1 g/l transferrin, 0.25 g/l insulin, 0.3 g/l L-glutamine, and 10% fetal bovine serum) was replaced with growth media (MEM containing 5 g/l glucose, 0.2 g/l NaHCO$_3$, 0.1 g/l transferrin, 0.3 g/l L-glutamine, 5% fetal bovine serum, 2% B-27 supplement, and 2 µM cytosine arabinoside) 2 days after plating. At 6 days *in vitro* (DIV), neurons were transduced with recombinant lentiviruses expressing myc-tagged Munc18-1 variants and cre recombinase. Wells receiving control, wild-type or mutant lentivirus, as well as neurons treated or not treated with compound, were randomized. Whenever possible, investigators were blinded toward the genotype and treatment group. Neurons were harvested or used for experiments as indicated at 13 DIV.

### Neuronal activity

Mouse cortical neurons were plated in 48-well CytoView MEA plates (Axion BioSystems) at a density of 160,000 cells per 10 µl containing 20 µg/ml laminin and transduced as above. Media was changed at DIV 12 and baseline spontaneous activity was recorded at DIV 14 for 15 min using the Maestro Pro MEA System (Axion BioSystems). Compounds or vehicle were added after baseline reading and recordings were repeated 48 h after compound addition. Extracellular voltage was recorded at each electrode with a sampling rate of 12.5 kHz. Spikes were identified from the raw signals using a detection threshold set independently for each electrode of ± 6x the standard deviation of the noise (AxIS Navigator software, Axion Biosystems). Electrodes with at least five spikes/min were considered active electrodes. The Neural Metric Tool (Axion BioSystems) was used for analysis of neuronal firing properties.

### Expression vectors

Full-length human Munc18-1b cDNA was inserted into pGEX-KG, pCMV5 or lentiviral vector FUW, containing an N-terminal myc-tag and a two amino acid linker, resulting in the following N-terminal sequence (EQKLISEEDL-GG). Mutant Munc18-1b constructs were generated by site-specific mutagenesis, according to the protocol of the manufacturer (Stratagene). N-terminally HA-tagged fragments of rat syntaxin-1 cDNA were amplified using PCR and inserted into pCMV5.

### Transfection of HEK293T cells

Cells were transfected with cDNA using calcium phosphate produced in house: 1 h prior to transfection, 25 µM chloroquine in fresh media was added. DNA was incubated for 1 min at room temperature in 100 mM CaCl$_2$ and 1x HBS (25 mM HEPES pH 7.05, 140 mM NaCl, and 0.75 mM Na$_2$HPO$_4$), and the transfection mix was then slowly added to the cells. Medium was replaced with fresh medium after 6 h. Cells were harvested or used for experiments as indicated 2 days after transfection. For lentivirus production, HEK293T cells were transfected with equimolar amounts of lentiviral vector FUW containing myc-tagged Munc18-1, pMD2-G-VSVg, pMDLg/pRRE, and pRSV-Rev. Medium containing the viral particles was collected 48 h later and centrifuged for 10 min at 2,000 rpm to remove cellular debris. Viral particles were subsequently concentrated tenfold by centrifugation.

### Total protein levels

Seven days after transduction of primary neurons, compounds were added at the indicated concentrations. Forty-eight hours later, cells were washed twice in PBS containing 1 mM MgCl$_2$ and solubilized in 2x Laemmli sample buffer containing 31 mg/ml DTT. Samples were then sonicated and boiled for 10 min at 100°C before separation by SDS–PAGE.

### Triton X-100 solubilization assay

Seven days after transduction of primary neurons, compounds were added at the indicated concentrations. Forty-eight hours later, cells were washed twice with PBS containing 1 mM MgCl$_2$ and removed from the dish using PBS. Cells were pelleted by centrifugation (5 min at 500 g$_{av}$) and were solubilized in 0.1% Triton X-100 in PBS supplemented with protease inhibitors for 1 h at 4°C under constant agitation. Insoluble material was pelleted by centrifugation (10 min at 13,000 g$_{av}$ and 4°C), the Triton X-100 soluble supernatant was transferred to a fresh tube, and the pellet was adjusted to the same volume with PBS. Both fractions were supplemented with 5x Laemmli sample buffer containing 77 mg/ml DTT and sonicated before separating equal volumes by SDS–PAGE.

### Antibody uptake assay

Cells were equilibrated for 10 min at room temperature in Krebs–Ringer solution (128 mM NaCl, 25 mM HEPES, 4.8 mM KCl, 1.3 mM CaCl$_2$, 1.2 mM MgSO$_4$, 1.2 mM KH$_2$/K$_2$HPO$_4$ [pH 7.4], 5.6 % glucose, pH 7.4). Medium was then replaced with Krebs–Ringer solution containing 55 mM KCl (and a corresponding reduction in NaCl) for 20 min at room temperature, containing 1:50 dilution of lumenal synaptotagmin-1 antibody. Cells were washed three times for 1 min with Krebs–Ringer solution and were then fixed with 4% PFA in PBS, and incubated with secondary antibody and DAPI. Neurons were imaged on an Eclipse 80i upright fluorescence microscope (Nikon) at same fluorescence intensity settings. For each image, distribution of the intensity of 300 pixels of dendrites and axons was analyzed, subtracted by average pixel intensity of the background, using ImageJ (NIH). Data were grouped into bins of five and number of pixels in each pixel intensity group was plotted.

### SynaptopHluorin experiments

Mouse cortical neurons were infected as described above and co-infected with lentiviral vectors expressing synaptophysin-pHluorin (SypHy) and TdTomato for neuronal identification. Forty-eight hours after compound addition, cells were equilibrated for 3 h at 37°C in Krebs–Ringer solution (128 mM NaCl, 25 mM HEPES, 4.8 mM KCl, 1.3 mM CaCl$_2$, 1.2 mM MgSO$_4$, 1.2 mM KH$_2$/K$_2$HPO$_4$ [pH 7.4], 5.6 % glucose, pH 7.4). SypHy fluorescence was excited using a 470 nm LED, and the change in fluorescence was monitored at 40x magnification using time-lapse recording with 600-ms intervals on an Eclipse TS2-FL inverted fluorescence microscope (Nikon) equipped with a Zyla Plus sCMOS 4.2 MP camera (Andor). Several frames were collected before the stimulus to determine the baseline SypHy fluorescence in cells, and exocytosis was then triggered with 55 mM KCl in Krebs-Ringer solution while recording. The

fluorescence intensity of SypHy pre- and post-stimulus was analyzed using Elements BR Analysis 5.21.02 Software (Nikon). The regions of interests were selected around the spatially resolved SypHy-positive puncta corresponding to synaptic terminals. Note that compound 10 is fluorescent at a similar wavelength than pHluorin so experiments at concentrations higher than 1 μM were not possible.

### Recombinant protein expression and purification

All proteins were expressed in bacteria (BL21 strain) as GST fusion proteins in modified pGEX-KG vectors (GE Healthcare) using auto-induction (Studier, 2005). Bacteria were harvested by centrifugation for 20 min at 3,500 rpm and 4°C, and pellets were resuspended in solubilization buffer (10 mM phosphate buffer pH 7.4, 500 mM NaCl, 5 mM DTT, 0.5% Triton X-100, 1 mM EDTA, 100 units DNase I, 1 mM $MgCl_2$, 5 mg/ml lysozyme, 1 mM PMSF, and an EDTA-free protease inhibitor cocktail (Roche)) and rotated for 1h at 4°C. Insoluble material was removed by centrifugation for 10 min at 12,000 $g_{av}$ and 4°C. Proteins were affinity-purified using glutathione sepharose bead (GE Healthcare) incubation for 2 h at 4°C, followed by extensive washing (10 mM phosphate buffer pH 7.4, 500 mM NaCl, 5 mM DTT, 15% glycerol) and elution for 1h at 4°C (5 mM reduced glutathione in 10 mM phosphate buffer pH 7.4, 300 mM NaCl, 5 mM DTT, 20% glycerol). Proteins were dialyzed overnight against elution buffer without reduced glutathione.

### *In vitro* compound binding

10 μg GST or GST-Munc18-1 WT, R406H, or G544D were immobilized on glutathione sepharose beads and incubated with 20 μM compound 9, 10, or 13 in PBS for 2h at 4°C in a 96 well plate. Beads were washed 3x with PBS, and fluorescence of protein-bound compound was measured using a Synergy H1 Hybrid Reader (BioTek). Excitation and emission wavelengths for the three compounds were: 350 $nm_{exc}$ and 450 $nm_{em}$ for compound 9, 475 $nm_{exc}$ and 530 $nm_{em}$ for compound 10, and 325 $nm_{exc}$ and 455 $nm_{em}$ for compound 13.

### Quantitation of average protein stability and function

Average stability was calculated from data obtained for total protein levels (percentage of DMSO; Appendix Fig S2 and Fig 3) and *C. elegans* heat shock paralysis (percentage of DMSO, set to 100%. Improvement in heat shock paralysis as increase from 100%; Fig 8). Average function was calculated from data obtained for spontaneous neural firing in MEA plate assay (percentage of DMSO; Fig 5 and Appendix Fig S4) and evoked neurotransmission in antibody uptake and synaptopHluorin fluorescence (percentage of DMSO; Figs 6 and 7).

### Limited proteolysis

Recombinant Munc18-1 (0.05 μg/μl) was incubated with or without 20 μM compound for 2 h at 4°C before addition of 0.001, 0.005, 0.01, 0.05, or 0.1 μg/μl trypsin on ice for 5 min. Tryptic digestion was immediately stopped by addition of 5x Laemmli sample buffer containing 77 mg/ml DTT and boiling for 10 min at 100°C.

### Quantitative immunoblotting

Protein samples were separated by SDS–PAGE and transferred onto nitrocellulose membranes. Blots were blocked in Tris-buffered saline (TBS) containing 0.1% Tween-20 (TBS-T) containing 5% fat-free milk for 30 min at room temperature. The blocked membrane was incubated overnight in PBS containing 1% BSA and 0.2% $NaN_3$ and the primary antibody. The blots were then washed twice in TBS-T containing 5% fat-free milk, then incubated for 1 h in the same buffer containing secondary antibody at room temperature. Blots were then washed 3 × in TBS-T, twice in water, and then dried in the dark. Blots were imaged using a LI-COR Odyssey CLx, and images were analyzed using ImageStudioLite (LI-COR).

### Antibodies

β-actin (A1978, Sigma, 1:1,000), GAPDH (DSHB-hGAPDH-2G7, DSHB, 1:500; G-9, Santa Cruz, 1:1,000), GFP (632381, Takara Bio Clontech, 1:2,000), Munc18-1 (610337, BD Biosciences, 1:1,000), myc (9E10, DSHB or C3956, Sigma, 1:500), synaptotagmin-1 (105221, Synaptic Systems, 1:50), Syntaxin-1 (HPC-1, Santa Cruz, 1:1,000). The monoclonal antibodies GAPDH (DSHB-hGAPDH-2G7) and myc (9E10) developed by DSHB and J.M. Bishop, respectively, were obtained from the Developmental Studies Hybridoma Bank, created by the NICHD of the NIH and maintained at The University of Iowa, Department of Biology, Iowa City, IA 52242.

### Quantification and statistical analysis

Sample sizes were chosen based on preliminary experiments or similar studies performed in the past. For quantification of immunoblots, a minimum of five independent experiments were performed. For quantification of immunofluorescence microscopy images, 5–10 neurons were analyzed regarding pixel intensity for the antibody uptake assay for each *n*. To ensure reliable quantification across samples and images, images were recorded under the same microscope settings (objective lens and illumination intensity). Merged images were created using Photoshop (Adobe) and were analyzed using ImageJ (NIH) or Image Studio (LI-COR). For quantification of *C. elegans* behavior, 10 animals were tested for each experiment, and at least 4 independent experiments were performed. No samples or animals were excluded from the analysis, and quantifications were performed blindly. Wells receiving control, wild-type or mutant lentivirus, as well as neurons treated or not treated with compound, were randomized. Whenever possible, investigators were blinded toward the genotype and treatment group. All data are presented as the mean ± SEM and represent a minimum of three independent experiments. Statistical parameters, including statistical analysis, significance, *P* value, and *n* value, are reported in each figure legend and/or in Appendix Table S1. Statistical analyses were performed using Prism 7 Software (GraphPad). For statistical comparison of two groups, either one-way ANOVA or two-way ANOVA was performed followed by Bonferroni or Dunnett's *post hoc* multiple comparisons tests, as indicated in the figure legends. Data were tested for normality, both by formal tests for normality (Kolmogorov–Smirnov, Shapiro–Wilk test, D'Agostino–Pearson test, and/or Anderson–Darling test), as well as visual inspection of Q–

## The paper explained

### Problem

Munc18-1/STXBP1 is essential for neuronal communication. Dominant mutations in Munc18-1 are linked to various severe epileptic encephalopathies and neurodevelopmental disorders. Although the molecular disease mechanisms underlying these syndromes are not yet fully understood, both haploinsufficiency and a dominant negative mechanism have been proposed, implying that 50% Munc18-1 expression is insufficient for normal function. Treatments are currently symptom-based and limited to the seizures associated with these syndromes, but do not modify the course of the disease and do not work for the majority of patients. Therefore, development of a Munc18-1-targeted therapy is necessary to treat the diverse and wide-ranging symptoms of Munc18-1 encephalopathies.

### Results

We performed a structure-based *in silico* screen of > 250,000 compounds from a diverse chemical library against the known structure of Munc18-1 and selected 17 compounds with the highest docking scores for further *in vitro* and *in vivo* validation. Of these, we identified three compounds that boosted protein levels of mutant Munc18-1, which is prone to rapid degradation and/or aggregation. Importantly, out of these three, one compounds elevated levels of WT Munc18-1 as well. The three compounds bound directly to Munc18-1 and two compounds functionally rescued the synaptic deficits seen in mutant Munc18-1 neurons in multiple assays. Finally, we demonstrate that the two compounds ameliorate mutant protein aggregation and synaptic dysfunction *in vivo*, using *C. elegans* models.

### Impact

We have identified two compounds that rescue not only the molecular deficits of mutant Munc18-1, but that also restore synaptic dysfunction, providing the first molecularly targeted, and possibly disease-modifying treatment strategy for Munc18-1 encephalopathies, which goes beyond the currently available single symptom-based treatment for epilepsy.

Q plots. If an abnormal distribution of data was noted, a non-parametric test was used followed by the appropriate multiple comparisons test, where applicable, and is noted in Appendix Table S1. A value of $P < 0.05$ was considered statistically significant.

## Data availability

This study includes no data deposited in external repositories.

**Expanded View** for this article is available online.

## Acknowledgements

We thank Dr. Thomas C. Südhof for providing antibodies and Dr. Matthijs Verhage for providing the conditional Munc18-1 knockout mice. This work was supported by T32GM007739 (Weill Cornell/Rockefeller/Sloan Kettering Tri-Institutional MD PhD Program for D.A.), 1F30HD100096-01A1 (to D.A.), the Alzheimer's Association (NIRG-15-363678 to M.S.), AFAR (M.S.), the NIH (1R01-AG052505 and 1R01-NS095988 to M.S.; R01-NS102181 and R01-NS113960 to J.B.), the Epilepsy Foundation & American Epilepsy Society (J.B.), the Leon Levy Foundation (J.B.), and the Sanofi Innovation Awards Program (J.B.).

## Author contributions

Study design, experiments, and data analysis: DA, NGLG, YN, MS, and JB; Designing of *in silico* studies: AD and GAP; Manuscript writing: DA and JB; All authors: Final manuscript discussion and comment.

## Conflict of interest

The authors declare that they have no conflict of interest.

## For more information

i  STXBP1 Foundation: https://www.stxbp1disorders.org/
ii  Author's website: https://www.burrelab.com

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
