## [Review Process File · EMBO Molecular Medicine]

Targeted stabilization of Munc18-1 function via pharmacological chaperones

Debra Abramov, Noah Guiberson, Andrew Daab, Yoonmi Na, Gregory Petsko, Manu Sharma, and Jacqueline Burre

DOI: [10.15252/emmm.202012354](https://doi.org/10.15252/emmm.202012354)

Corresponding author(s): Jacqueline Burre (jab2058@med.cornell.edu) , Debra Abramov (dea2015@med.cornell.edu)

Review Timeline:

Submission Date:	17th Mar 20
Editorial Decision:	23rd Apr 20
Revision Received:	7th Oct 20
Editorial Decision:	26th Oct 20
Revision Received:	1st Nov 20
Accepted:	11th Nov 20

Editor: Jingyi Hou

Transaction Report:

Thank you for the submission of your manuscript to EMBO Molecular Medicine. We have now received feedback from the three referees who agreed to evaluate your manuscript. As you will see from the reports below, while referee #2 is overall positive, both referee #1 and #3 are more reserved and raise substantial concerns about your work, which should be convincingly addressed in a major revision of the presented manuscript.

In particular, during our cross-commenting process (in which the referees are given the chance to make additional comments, including on each other's reports), referee #1 added "As mentioned in my report, the alpha-screen would seem a more appropriate and accepted technique to screen candidates. I was worried about the level of variability in the loading controls of the WB and was fairly unconvinced by the results. In essence, this would mean a major change in the paper with a completely new approach to probe for candidates and then use rounds of derivatizations to generate sub-micromolar affinity molecules. This is needed because the rescue of the paralytic phenotype in *c. elegans* is minor to say the least, despite high concentrations of the top compounds used. I agree with referee #3, the stats are not appropriate. Analysis of the variation in intensity of the loading controls should show that their significant data are within the noise and therefore very little can be drawn from this type of analysis." Referee #3 added "I agree that better compounds are needed to be really useful. But without better compounds, the authors need to demonstrate that the current compounds REALLY have an effect (albeit small). This is in line with the assessment that the WB results are not convincing. In fact I was surprised to see the WB images and the "significant" plots, and started wondering how this can be and realized that the authors used t-test for all pairwise comparisons instead of one-way anova."

The main critical point is about the validity of the in-silico screening and the Western blot results. The second critical point is about the statistical tests and the subtlety of the effects. We would not ask you to use a completely different screening approach, as this would likely not be feasible in a reasonable timeframe. However, the rationale for the current in-silico screening method should be more clearly stated and its potential limitations should be discussed. The other concerns with regards to the Western Blot and stats must be convincingly addressed.

I would like to give you the opportunity to revise your manuscript, with the understanding that the referee concerns must be convincingly addressed and that acceptance of the manuscript would entail a second round of review. Please note that EMBO Molecular Medicine strongly supports a single round of revision and that, as acceptance or rejection of the manuscript will depend on another round of review, your responses should be as complete as possible.

***** Reviewer's comments *****

Referee #1 (Comments on Novelty/Model System for Author):

As indicated in my report, I was not convinced by their screening method. Further the WB analysis was also very variable and seems inappropriate for efficiently testing compounds. I suggested an alpha-screen which has been used in the past successfully in screening compounds. Not surprisingly the best compounds had little to no rescuing effect on the paralytic phenotype of their *c. elegans* model.

Referee #1 (Remarks for Author):

In the study the authors have used a compound library approach to find molecular chaperone that can rescue disease-causing mutations in Munc18-1, also called syntaxin-binding protein-1 and responsible for a variety of conditions ranging from Early Infantile Epileptic encephalopathy, autism, and intellectual disability. They first look that the ability of syntaxin1A expression to rescue Munc18-1 from aggregating. They used an in-silico molecular docking methods to select compounds from a library for their ability to bind several areas of Munc18 and established an in vitro assay based on western blotting analysis to test the best hits. They finally tested the

therapeutically promising molecules to assess their rescuing ability in hippocampal neurons and in *c. elegans* expressing Munc18-1 variants.

I am not overly positive about this manuscript because of major issues with screening and testing of compounds as detailed below. Not surprisingly, the compounds selected have a very minor rescuing effect on their *c. elegans* small animal model. Overall, this study seems preliminary.

1- I am unsure about the validity of the in-silico screening platform used. It seems more appropriate to use a screening method based on binding to test the compound library and then check where the molecules of interest are likely to bind. Alpha-screens amongst other type of basic assays, have been very useful in other studies.

2-The western blotting assay, looking at the rescue of soluble expression of Munc18-1 is unconvincing with Munc18-1 bands looking fairly homogeneously distributed and some worrying level of variability in the loading controls. A more quantitative assay, such as alpha-screen will be needed to test the library compounds.

3- Once a candidate has been selected, rounds of derivatization must be carried out to generate sub-micromolar affinity molecules.

4- The assay used to test the recycling of synaptic vesicles is unreliable and has long been replaced in the field by more accurate and appropriate methods such as the use of pHluorin and other pH-sensitive probes. These should be used to assess the effect of the best candidates.

5- Finally, the paralytic phenotype in *c. elegans* is only slightly delayed by a few minutes in rescue conditions, despite high concentrations of the top compounds suggesting that the therapeutic rescue is not working properly.

Referee #2 (Comments on Novelty/Model System for Author):

Treating patients with heterozygous Munc18-1 mutations is a major medical need. This paper describes a potential avenue to achieving this goal.

Referee #2 (Remarks for Author):

This study reports a major advance in developing drugs to help re-nature mutant Munc18-1 in patients with heterozygous Munc18-1 mutations. I think the results are important, well done, and should be published. I won't dwell further on what I think is important here because it seems obvious to me. However, I have three areas of suggested improvements that would further elevate the level of this paper:

a. In Figure 6, the dose-response effect of the compounds is difficult to assess. The authors should test more doses. Moreover, the authors should test the effect of the compounds on WT neurons. Is compound 10 an inhibitor of Munc18-1 function?

b. In Figure 7, the authors may consider testing WT worms and including a 10 min time point. They may have missed the most important effect.

c. In Figure 8, the authors should measure actual binding of the compounds to both WT and mutant Munc18-1. Given their structure, the compounds are likely fluorescent. A fluorescence polarization assay should be an easy way to measure affinities. Better even would be ITC.

In addition to these major suggestions, I think the authors could use less jargon - what are 'primary' neurons as opposed to 'secondary' neurons?

Overall, this is a very nice paper that deserves publication. I hope my comments will be helpful.

Referee #3 (Comments on Novelty/Model System for Author):

Both cultured cells and worms are appropriate.

Referee #3 (Remarks for Author):

Heterozygous mutations of Munc18-1 cause a number of neurological conditions. Previous studies including the work from the Burre lab showed that some Munc18-1 missense mutations lead to destabilization and aggregation of the protein. In this manuscript, Abramov and colleagues performed an in-silico screen to find small molecules that bind to Munc18-1. They identified several compounds that can stabilize the Munc18-1 G544D mutant protein and reduce the neuronal dysfunction. Thus, these small molecules can potentially be used to treat Munc18-1 associated conditions. The in-silico screen was productive and several phenotypic characterizations in vitro and in vivo yielded interesting results. However, there are a number of concerns that should be addressed before publication of this manuscript.

Major concerns:

1. The inappropriate use of statistical tests precludes interpretation of the results. The authors used t-test for all multi-group comparisons except fig7a. The appropriate test should be one-way ANOVA or a non-parametric test such as Kruskal-Wallis test, depending on the distribution of the data. These include fig1, fig3, fig4, fig5, fig6, fig7b, fig8, suppl fig1, suppl fig2, and suppl fig4. Many experiments have a small effect size and it's possible that the reported significance simply resulted from the inappropriate t-test.
2. The authors argue that the binding sites of the compounds are structurally distinct from the G544D mutation, thus the compounds should work for other missense mutations. However, as the authors showed, the compounds didn't increase the WT protein level, except compound 9 having an effect at one concentration. This result actually suggests that the effect of these compounds is specific to the G544D mutation, may not translate to other mutations, and will not work for those heterozygous deletion cases.
3. In fig5, compounds 9 and 10 at both 1uM and 5 uM caused a dramatic enhancement of synaptotagmin-1 antibody uptake, which presumably reflects an increase of synaptic transmission. However, compound 9 at 5 uM and compound 10 at 1 uM and 5 uM didn't rescue the neuronal firing at all (fig6). Compound 13 at 1 uM and 20 uM barely enhanced synaptotagmin-1 antibody uptake, yet caused a large increase in neuronal firing at 5 uM. Without a better understanding of these inconsistent results, it is difficult to draw conclusions about the effects of these compounds.
4. The results in Fig7d,e should to be quantified.
5. The title, abstract, and text claim that the identified compounds are target-specific. However, the manuscript only showed the interaction of these compounds with Munc18-1, but no data about the

specificity of these compounds. Without testing if these compounds have any other targets, the authors should not make such a claim.

Minor points:

1. Fig5a: the scale bar should be μm , not μM .
2. The n numbers should be specified for each group, not a range for all groups together.

We thank the reviewers for their insightful comments. As described below, we have added not only a second Munc18-1 mutant, but also a number of additional experiments to expand and strengthen our findings. Moreover, we made significant changes in the text in order to respond to the reviewers' and editor's suggestions. Please note, that for a better flow of the manuscript, we have changed the order of figures.

Briefly, the following new data were added to the revised manuscript:

- 1) Extension of our findings to an additional Munc18-1 missense mutation, R406H, in all of our model systems (Figures 3-8), demonstrating that the effect of our identified compounds is not restricted to G544D.
- 2) Demonstration of direct binding of our compounds to both wild-type and two mutant versions of Munc18-1 (Figure 4C) which has replaced the previous thermofluor assay.
- 3) Re-analysis of our data using appropriate statistical methods. Please see Appendix Table S1 for details on testing of normality, statistical tests used, and exact n and p values for each figure panel.
- 4) Additional experiments and inclusion of more representative immunoblots (Figures 3 & 4).
- 5) Inclusion of additional concentrations of compounds (Figures 3, 5, 6 and 8).
- 6) Testing of the effect of our compounds on neurotransmitter release using a synaptophysin-pHluorin assay (new Figure 7) which complements our antibody uptake studies (Figure 6) and MEA data (Figure 5).
- 7) New *C. elegans* heat shock experiments, with addition of earlier paralysis time points, dose-response curves, as well as wild-type and R405H worm strains (Figure 8).
- 8) Addition of a justification for our *in silico* screen, highlighting advances and disadvantages over physical screens.

We hope that the reviewers and editor will find these substantial changes convincing, and will deem the paper acceptable for publication.

Reviewers' comments:

Referee #1 (Comments on Novelty/Model System for Author):

As indicated in my report, I was not convinced by their screening method. Further the WB analysis was also very variable and seems inappropriate for efficiently testing compounds. I suggested an alpha-screen which has been used in the past successfully in screening compounds. Not surprisingly the best compounds had little to no rescuing effect on the paralytic phenotype of their *c. elegans* model.

We respectfully disagree regarding the screening method. While the chosen screening method dictates to some extent what compounds will be found, of greater importance is what library of compounds is screened. Our lab does not have access to automated high-throughput screening of millions of compounds, so a sensible strategy was to screen a vast library of compound structures *in silico*, with the hope that the docking will winnow the vast library down to a hundred or fewer potential compounds that can actually be obtained and screened experimentally.

The advantage of the *in silico* technique is that it increases the hit rate of a physical screen from less than 0.01%, which is typical when a totally random library is screened, to 1-10% by biasing the experiment to compounds that are predicted to be likely to bind. I.e., it produces a list of likely binders that is manageable to screen physically and will nearly always give a few hits, whereas otherwise, hundreds of thousands of compounds would have to be physically screened in an alpha screen to get a few hits.

The weakness of the *in silico* method, admittedly, is that the energy functions used to assess interactions are crude at best, do not take entropy into account, and cannot be used even to estimate the likely strength of binding. False positives may abound, and we know nothing about false negatives, i.e. nobody has gone to the trouble to see how many compounds at the bottom might bind anyway. So overall, we do not know what we missed.

While we agree that the method is far from perfect, like any other method, there are many examples in the literature of its successful application. We have now included a discussion of advantages and disadvantages of our screening method.

Regarding the second comment about WB analysis: With inclusion of the additional Munc18-1 mutant R406H, we have significantly increased our n for each experiment, reducing variability in our western blot analyses, also demonstrated by better and more consistent loading controls (see Figures 3 and S2). Yet, we would like to point out that there will always be inherent differences in loading, not only because so many samples are run side-by-side, but also because each band represents the signal from one well of cultured neurons which vary in speed of growth and spread from well to well. To account for this variation, all our data are always normalized to a loading control. Please also note that for some compounds, variability in WB results is due to reduced neuronal survival (especially for 250 μ M compound; Figure 3 and S2).

Referee #1 (Remarks for Author):

In the study the authors have used a compound library approach to find molecular chaperone that can rescue disease-causing mutations in Munc18-1, also called syntaxin-binding protein-1 and responsible for a variety of conditions ranging from Early Infantile Epileptic encephalopathy, autism, and intellectual disability. They first look that the ability of syntaxin1A expression to rescue Munc18-1 from aggregating. They used an in-silico molecular docking methods to select compounds from a library for their ability to bind several areas of Munc18 and established an in

vitro assay based on western blotting analysis to test the best hits. They finally tested the therapeutically promising molecules to assess their rescuing ability in hippocampal neurons and in *C. elegans* expressing Munc18-1 variants.

I am not overly positive about this manuscript because of major issues with screening and testing of compounds as detailed below. Not surprisingly, the compounds selected have a very minor rescuing effect on their *C. elegans* small animal model. Overall, this study seems preliminary.

Please see above and below for specific responses to these comments.

1- I am unsure about the validity of the in-silico screening platform used. It seems more appropriate to use a screening method based on binding to test the compound library and then check where the molecules of interest are likely to bind. Alpha-screens amongst other type of basic assays, have been very useful in other studies.

Please see above under “Comments on Novelty/Model System for Author” for our response to this comment.

2-The western blotting assay, looking at the rescue of soluble expression of Munc18-1 is unconvincing with Munc18-1 bands looking fairly homogenously distributed and some worrying level of variability in the loading controls. A more quantitative assay, such as alpha-screen will be needed to test the library compounds.

Regarding the suggested alpha-screen, please see our comments above under “Comments on Novelty/Model System for Author”.

We have now removed the detergent solubility assay because both, R406H and G544D are so insoluble that due to their already low expression levels, background contributes largely to variation. Instead, we have now quantified our solubility data in *C. elegans*, which has a much higher signal to noise ratio (Figure 8M-P).

3- Once a candidate has been selected, rounds of derivatization must be carried out to generate sub-micromolar affinity molecules.

Please see above under “Comments on Novelty/Model System for Author” for our comments regarding the choice of screening method. While rounds of derivatization are desirable for our lead compounds, this is beyond the scope of this manuscript. Derivatization of our lead compounds is something we plan to do in the future, in combination with developing a mouse model of disease, to simultaneously analyze toxicity, pharmacology, and blood-brain barrier permeability.

4- The assay used to test the recycling of synaptic vesicles is unreliable and has long been replaced in the field by more accurate and appropriate methods such as the use of pHluorin and other pH-sensitive probes. These should be used to assess the effect of the best candidates.

We would first like to point out that many laboratories still rely on antibody uptake assays, including prominent scientists in the field such as Drs. Kavalali, Rizzoli, Chapman. For example, please see:

- Richter KN, Patzelt C, Phan NTN, Rizzoli SO. *Scientific Reports* (2019) 91: 9231.
- Lindhout FW, Cao Y, Kevenaer JT, Bodzeta A, Stucchi R, Boumpoutsari MM, Katrukha EA, Altelaar M, MacGillavry HD, Hoogenraad CC. *EMBO J* (2019): e101345.
- Truckenbrodt S, Viplav A, Jähne S, Vogts A, Denker A, Wildhagen H, Fornasiero EF, Rizzoli SO. *EMBO J* (2018).
- Thalhammer A, Contestabile A, Ermolyuk YS, Ng T, Volynski KE, Soong TW, Goda Y, Cingolani LA. *Cell reports* (2017) 202: 333-343.
- Martineau M, Somasundaram A, Grimm JB, Gruber TD, Choquet D, Taraska JW, Lavis LD, Perrais D. *Nature communications* (2017) 81: 1412.
- Ramirez DMO, Crawford DC, Chanaday NL, Trauterman B, Monteggia LM, Kavalali ET. *J Neurosci* (2017) 3726: 6224-6230.
- Hayashi Y, Nishimune H, Hozumi K, Saga Y, Harada A, Yuzaki M, Iwatsubo T, Kopan R, Tomita T. *Scientific reports* (2016) 6: 23969.

However, we do agree that pHluorin-based assays have certain advantages and have now included measurements of neurotransmitter release using synaptophysin-pHluorin (see new Figure 7). Overall, the data of our assays largely agree (please compare Figures 5, 6 and 7).

5- Finally, the paralytic phenotype in *c. elegans* is only slightly delayed by a few minutes in rescue conditions, despite high concentrations of the top compounds suggesting that the therapeutic rescue is not working properly.

We have now added wild-type worms to our readout, which demonstrates a significant rescue of mutant worms with our compounds (Figure 8). In addition, reviewer 2 made the suggestion to include earlier time points in our measurements, which in fact reveals the rescuing phenotype much better. Note that we have also included a second mutant worm strain carrying the R405H mutation in UNC18 (the R406H homolog of human Munc18-1), which shows a similar trend compared to G544D UNC18.

Referee #2 (Comments on Novelty/Model System for Author):

Treating patients with heterozygous Munc18-1 mutations is a major medical need. This paper describes a potential avenue to achieving this goal.

We thank the reviewer for this positive evaluation.

Referee #2 (Remarks for Author):

This study reports a major advance in developing drugs to help re-nature mutant Munc18-1 in patients with heterozygous Munc18-1 mutations. I think the results are important, well done, and should be published. I won't dwell further on what I think is important here because it seems obvious to me.

Again, we thank the reviewer for these very positive comments.

However, I have three areas of suggested improvements that would further elevate the level of this paper:

a. In Figure 6, the dose-response effect of the compounds is difficult to assess. The authors should test more doses. Moreover, the authors should test the effect of the compounds on WT neurons. Is compound 10 an inhibitor of Munc18-1 function?

We agree and have now included a wider range of concentrations (now Figures 3 & 5), in addition to measuring the effect of our compounds on a second missense mutation, R406H. Note, that while 0.25 μ M seems to work well for some compounds, at 250 μ M neurons are not surviving well.

Regarding a dose-response curve, it appears that compounds 9 and 13 are active over a wide range, while compound 10 can only be used at lower concentration, due to its toxic effects at higher concentrations (see loading controls in Figures 3 and S2). The reviewer is correct to notice the silencing effect of compound 10 on spontaneous release in the MEA assay at higher concentrations, compared to the structurally related compound 13. This is likely due to the toxic effects on neurons. Yet, under evoked release, when we measured synaptic vesicle cycling using an antibody uptake assay, or neurotransmitter release using synaptophysin-pHluorin, compound 10 boosted uptake significantly at lower dose. We conclude that compound 10 is toxic at higher dose, and similarly effective as compound 13 at lower dose.

b. In Figure 7, the authors may consider testing WT worms and including a 10 min time point. They may have missed the most important effect.

We thank the reviewer for this suggestion and have now repeated this assay to include more monitoring points in the beginning in addition to testing a separate worm strain with a different missense mutation (R405H, the worm equivalent of human R406H Munc18-1) as well as wild-type worms at various compound concentrations (now Figure 8).

c. In Figure 8, the authors should measure actual binding of the compounds to both WT and mutant Munc18-1. Given their structure, the compounds are likely fluorescent. A fluorescence polarization assay should be an easy way to measure affinities. Better even would be ITC.

An excellent idea! We have found all three compounds to be fluorescent. Regretfully, nobody at Weill Cornell or in close vicinity has a plate reader with the capability to measure fluorescence polarization, and ITC was not possible due to COVID-19 related shutdown of the cores. Yet, we were able to measure direct binding of compounds to immobilized Munc18-1 variants where we

incubated bead-immobilized GST alone, GST-tagged wild-type, R406H or G544D with our compounds, and measured bead-bound fluorescence upon extensive washing. Our new data reveal that all three compounds directly bind to WT and mutant Munc18-1 (new Figure 4C).

In addition to these major suggestions, I think the authors could use less jargon - what are 'primary' neurons as opposed to 'secondary' neurons?

We have now changed this to “cortical neurons” or “neurons”. Primary neurons is a conventional term referring to neurons plated directly from animal brains (P0 pups), aiming to differentiate these neurons from heterologous cells that can be passaged in a dish.

Overall, this is a very nice paper that deserves publication. I hope my comments will be helpful.

We thank the reviewer very much for the positive evaluation and helpful comments which have significantly improved our manuscript.

Referee #3 (Comments on Novelty/Model System for Author):

Both cultured cells and worms are appropriate.

We thank the reviewer for this evaluation.

Referee #3 (Remarks for Author):

Heterozygous mutations of Munc18-1 cause a number of neurological conditions. Previous studies including the work from the Burre lab showed that some Munc18-1 missense mutations lead to destabilization and aggregation of the protein. In this manuscript, Abramov and colleagues performed an in-silico screen to find small molecules that bind to Munc18-1. They identified several compounds that can stabilize the Munc18-1 G544D mutant protein and reduce the neuronal dysfunction. Thus, these small molecules can potentially be used to treat Munc18-1 associated conditions. The in-silico screen was productive and several phenotypic characterizations in vitro and in vivo yielded interesting results.

We thank the reviewer for assessing our work productive and yielding interesting results.

However, there are a number of concerns that should be addressed before publication of this manuscript.

Major concerns:

1. The inappropriate use of statistical tests precludes interpretation of the results. The authors used t-test for all multi-group comparisons except fig7a. The appropriate test should be one-way ANOVA or a non-parametric test such as Kruskal-Wallis test, depending on the distribution of

the data. These include fig1, fig3, fig4, fig5, fig6, fig7b, fig8, suppl fig1, suppl fig2, and suppl fig4. Many experiments have a small effect size and it's possible that the reported significance simply resulted from the inappropriate t-test.

We apologize for this oversight. We have now re-analyzed all of our data using the appropriate statistical tests (see Appendix Table S1 for details on testing of normality, statistical tests used, and exact n and p values for each figure panel). Please note that due to addition of another missense mutation and additional compound concentrations, some of the assays were either re-run or re-done, adding more experimental data to the previous existing data.

2. The authors argue that the binding sites of the compounds are structurally distinct from the G544D mutation, thus the compounds should work for other missense mutations. However, as the authors showed, the compounds didn't increase the WT protein level, except compound 9 having an effect at one concentration. This result actually suggests that the effect of these compounds is specific to the G544D mutation, may not translate to other mutations, and will not work for those heterozygous deletion cases.

To test if the effect of our compounds is restricted to the G544D mutation, we have chosen to analyze a second missense mutation, R406H. The results of our readouts for both mutants are very similar (compare data in Figures 3-8). In addition, compound 9 and 13 increased WT levels in hemizygous neurons (Figure S2), a proper model for mutations other than missense mutations. These data demonstrate that our compounds are efficacious for at least two different missense mutations that are structurally distinct and that are in no close vicinity, and may work on WT in a hemizygous condition as well.

3. In fig5, compounds 9 and 10 at both 1uM and 5 uM caused a dramatic enhancement of synaptotagmin-1 antibody uptake, which presumably reflects an increase of synaptic transmission. However, compound 9 at 5 uM and compound 10 at 1 uM and 5 uM didn't rescue the neuronal firing at all (fig6). Compound 13 at 1 uM and 20 uM barely enhanced synaptotagmin-1 antibody uptake, yet caused a large increase in neuronal firing at 5 uM. Without a better understanding of these inconsistent results, it is difficult to draw conclusions about the effects of these compounds.

As suggested by reviewer 1, we have now added a pHluorin-based assay in addition to the MEA assay and antibody uptake assay. Like the antibody uptake assay, the pHluorin-based assay tests evoked neurotransmitter release, and the results of these two experiments are similar. Comparing the MEA data (spontaneous release) with the uptake and pHluorin data (evoked release), the reviewer is correct to note that spontaneous release is rescued less by the compounds. This discrepancy may be due to the inherent differences of the experiments. The MEA plate assay measures the mean firing frequency of all neurons, both active and inactive, within each well and will give the average of both. Both the pHluorin-based assay and the antibody uptake assay measure only the active cells, as these are the ones that will fire with stimulation and "light up." The differences of results seen in these experiments are thus likely due to this technical difference in the experiments, not due to the differences of compound effect on neuronal activity. We have included now a discussion of these findings in the manuscript.

4. The results in Fig7d,e should to be quantified.

Thank you for the suggestion. These data have been quantified now (new Figure 8P).

5. The title, abstract, and text claim that the identified compounds are target-specific. However, the manuscript only showed the interaction of these compounds with Munc18-1, but no data about the specificity of these compounds. Without testing if these compounds have any other targets, the authors should not make such a claim.

We completely agree and have modified the title, abstract and text accordingly.

Minor points:

1. Fig5a: the scale bar should be μm , not μM .

We have corrected this now.

2. The n numbers should be specified for each group, not a range for all groups together.

While we have kept the range in the figure legends for simplicity, details on statistical analyses, exact p and n values are provided now in Appendix Table S1.

Thank you for the submission of your revised manuscript to EMBO Molecular Medicine. We have now received the enclosed report from the referees who were asked to re-assess it. You will see from the comments below that the referees think that while the majority of their concerns have been addressed, several issues remain. We would not request additional experiments at this stage. However, the following referee comments still need to be addressed:

- Please fix the small inconsistencies commented by referee 1.
- Referee #3 still raised a couple of concerns with regards to the variable effect sizes in different assays, which need to be clarified and discussed in writing.

Please provide a point-by-point letter INCLUDING my comments and the reviewer's reports and your detailed responses to their comments (as Word file).

On a more editorial level, please do the following:

***** Reviewer's comments *****

Referee #1 (Comments on Novelty/Model System for Author):

The authors did a great job answering my queries and criticisms. My main worry was that the

rescue times were small but I did not take into account the actual lifespan of c elegans. By simply plotting the WT, one can see the proper extent of the the rescue which is significant. I think this is a good paper that certainly deserve publication.

Referee #1 (Remarks for Author):

The authors did a great job answering my queries and criticisms. My main worry was that the rescue times were small. Plotting the WT gives a much better indication of the level of rescue in c elegans. The pHluorin exps confirms their previous results which is reassuring.

There are still some small inconsistencies here and there that may require the authors attention. In particular Fig. 3 panel B, the numbering of the compounds does not fit with the bars which is a bit confusing.

Compounds 10? seems be have the potential to work at nM concentrations in Fig. 3 but not Fig. 7 (release experiment).

Referee #2 (Comments on Novelty/Model System for Author):

Munc18-1 mutations are a major cause of Ohtahara syndrome, and treating patients with such mutations would be enormously beneficial.

Referee #2 (Remarks for Author):

I am satisfied by the authors revisions, and recommend the paper for publication.

Referee #3 (Comments on Novelty/Model System for Author):

Both cultured cells and worms are appropriate.

Referee #3 (Remarks for Author):

The authors have addressed many of this reviewer's previous concerns and significantly improved the manuscript, particularly the statistics. However, with the proper statistical analyses, one main issue remains, that is the variable effects and effect sizes in different assays, which raises the concern about the reliability of the assays.

1. It's nice that the authors tested the compounds on another mutation (R406H) and hemizygous neurons. However, effects of these compounds varied among R406H, G544D, and hemizygous neurons. It's understandable that the statistical significance may not be reached for some concentrations, but the trends or the patterns of the effects are quite different. For example, compound 9 at lower concentrations had best effect on G544D level, but no effect on R406H and hemizygous neurons. Compound 13 had effects on G544D level at both low and high concentrations, but no effect on hemizygous neurons.

2. The authors argued that the difference among MEA, antibody uptake, and pHluorin assays is due to the measurement of both active and inactive neurons for MEA, but only the active neurons for antibody uptake and pHluorin. This argument doesn't make sense to the reviewer, unless the authors purposely selected subset of the so called "active neurons" for analysis in the antibody uptake and pHluorin experiments. Clearly the authors should be able to analyze all neurons including the inactive ones, as they did for the control condition (DMSO). This need to be clarified.

3. More importantly, the trends or the patterns of the effects in the functional assays often are different from those on the protein levels. One possible way to help visualizing the patterns of protein levels vs functional outcomes would be to make some correlation plots.

We thank the editor and reviewers again for their insightful comments. As described below, we have added a new Figure and rewritten parts of our manuscript to address the remaining concerns

***** Editor's comments *****

Thank you for the submission of your revised manuscript to EMBO Molecular Medicine. We have now received the enclosed report from the referees who were asked to re-assess it. You will see from the comments below that the referees think that while the majority of their concerns have been addressed, several issues remain. We would not request additional experiments at this stage. However, the following referee comments still need to be addressed:

-Please fix the small inconsistencies commented by referee 1.

This has been fixed now.

-Referee #3 still raised a couple of concerns with regards to the variable effect sizes in different assays, which need to be clarified and discussed in writing.

We have added a correlation plot (new Figure 9) and a paragraph to the discussion to clarify this issue.

***** Reviewer's comments *****

Referee #1 (Comments on Novelty/Model System for Author):

The authors did a great job answering my queries and criticisms. My main worry was that the rescue times were small but I did not take into account the actual lifespan of *c. elegans*. By simply plotting the WT, one can see the proper extent of the the rescue which is significant. I think this is a good paper that certainly deserve publication.

We thank the reviewer for this very positive evaluation.

Referee #1 (Remarks for Author):

The authors did a great job answering my queries and criticisms. My main worry was that the rescue times were small. Plotting the WT gives a much better indication of the level of rescue in *c. elegans*. The pHluorin exps confirms their previous results which is reassuring.

Again, we thank the reviewer for the positive evaluation.

There are still some small inconsistencies here and there that may require the authors attention. In particular Fig. 3 panel B, the numbering of the compounds does not fit with the bars which is a bit confusing.

Compounds 10? seems be have the potential to work at nM concentrations in Fig. 3 but not Fig. 7 (release experiment).

We thank the reviewer for pointing this out, and have fixed these inconsistencies. In addition, we have now included a correlation plot (new Figure 9) as suggested by reviewer 3, in addition to a paragraph in the discussion to clarify the effect of the compounds at each concentration regarding protein stability and functional rescue.

Referee #2 (Comments on Novelty/Model System for Author):

Munc18-1 mutations are a major cause of Ohtahara syndrome, and treating patients with such mutations would be enormously beneficial.

Referee #2 (Remarks for Author):

I am satisfied by the authors revisions, and recommend the paper for publication.

We thank the reviewer for recommending publication of our work.

Referee #3 (Comments on Novelty/Model System for Author):

Both cultured cells and worms are appropriate.

Referee #3 (Remarks for Author):

The authors have addressed many of this reviewer's previous concerns and significantly improved the manuscript, particularly the statistics. However, with the proper statistical analyses, one main issue remains, that is the variable effects and effect sizes in different assays, which raises the concern about the reliability of the assays.

1. It's nice that the authors tested the compounds on another mutation (R406H) and hemizygous neurons. However, effects of these compounds varied among R406H, G544D, and hemizygous neurons. It's understandable that the statistical significance may not be reached for some concentrations, but the trends or the patterns of the effects are quite different. For example, compound 9 at lower concentrations had best effect on G544D level, but no effect on R406H and hemizygous neurons. Compound 13 had effects on G544D level at both low and high concentrations, but no effect on hemizygous neurons.

We thank the reviewer for this comment. As suggested below under point 3, we have now included a correlation plot for G544D and R406H Munc18-1 that summarizes the effects of the compounds on protein stability and function (new Figure 9). The plot highlights that for both mutants, compounds 9 and 13 work at 1 μ M, 5 μ M and 20 μ M concentration, whereas compound 10 had only minor effects at 1 and 5 μ M. Please note that for both mutants, high concentrations of all compounds have negative effects on Munc18-1 function, despite an increase in protein levels. We speculate that this is due to off-target effects of the compounds that may impair overall neuronal function.

The reviewer is correct to point out that the effects of compounds 9 and 13 on both mutants differ slightly: compound 9 is more potent in stabilizing R406H Munc18-1, whereas compound 13 shows higher efficacy on G544D Munc18-1. We speculate that this may be due to the domain where the mutation is located and where the compounds bind. R406H and G544D are located in different domains, likely leading to differences in folding/misfolding. Stabilization of the area that is primarily destabilized is expected to have higher efficiency in stabilizing the entire protein, i.e. compound 9 for R406H, and compound 13 for G544D.

In contrast, hemizygous neurons express WT Munc18-1 which has no local folding deficit as mutant Munc18-1, but is overall metastable, and the effect of the compounds is likely due to the pathway to attain its native state.

Furthermore, although the effect of compounds 9 and 13 on hemizygous neurons are not significant by one-way ANOVA, they do have significant effects compared to DMSO: compound 13 at 5uM and 20uM increase WT protein level, as does compound 9 at 20uM.

We have included now a paragraph in the discussion section highlighting these findings.

2. The authors argued that the difference among MEA, antibody uptake, and pHluorin assays is due to the measurement of both active and inactive neurons for MEA, but only the active neurons for antibody uptake and pHluorin. This argument doesn't make sense to the reviewer, unless the authors purposely selected subset of the so called "active neurons" for analysis in the antibody uptake and pHluorin experiments. Clearly the authors should be able to analyze all neurons including the inactive ones, as they did for the control condition (DMSO). This need to be clarified.

We apologize for our unclear wording. The words "active" and "inactive" were used to draw a distinction between neurons that do and do not fire. They do not indicate which neurons were chosen by us for analysis in each assay. All three assays compare the results of the control condition (DMSO groups) to the treated condition (compound groups).

Because Munc18-1 is involved in both spontaneous and evoked neurotransmitter release, we performed assays to determine whether compounds improve either or both functions. In the case of evoked neurotransmission, we stimulate neurons by adding potassium in each treatment condition, and the activated synapses fluoresce - either due to antibody uptake or unquenching of synaptophluorin - and the amount of fluorescence is then compared between the compound-treated and DMSO-treated cells. In both cases, fluorescence only occurs if the synapse releases neurotransmitter.

The MEA assay measures spontaneous neural activity of the neuronal network in vicinity of electrodes (local field potentials) in each treatment condition, without stimulation. Geometric and electrical constraints largely limit its ability to detect action potentials only at the soma. Also, quick fluctuations in the potential difference caused by single action potentials are mostly undetectable, leaving only the slower fluctuations, composed of the more sustained currents in the network, typical of the somato-dendritic currents. This is very different from the single synapse-signals in our fluorescence assays. In WT neurons, most neurons fire spontaneously with a measurable frequency; however, far fewer mutant Munc18-1 neurons fire due to synaptic dysfunction and the mean firing frequency of local field potentials is far lower. Despite this lower signal, we did find that compounds 9 and 13 improved this spontaneous firing compared to DMSO-treated mutant neurons.

In the evoked neurotransmission experiments, the quantifiable signal comes from specific synapses that release neurotransmitter in mutant Munc18-1 neurons, i.e. synapses that do not release neurotransmitter do not show up and thus do not contribute to the measured signal. In the MEA assay, the mean firing frequency reported for each treatment condition is an average of spontaneous somato-dendritic currents (including those that do not result in downstream action potentials) of all neurons in the network, i.e. those which spontaneously fire and those which will not during the course of the experiment. We therefore believe the different results may reflect the technical differences between these assays. We have included both assays in the paper, given the importance of Munc18-1 in both forms of neurotransmitter release. Overall, these assays measure neuronal activity very differently, i.e. presynaptic release versus somato-

dendritic local field potentials, both of which are apparently affected by presynaptic release frequency, but may not follow a simple correlation.

This has also been clarified in the manuscript.

3. More importantly, the trends or the patterns of the effects in the functional assays often are different from those on the protein levels. One possible way to help visualizing the patterns of protein levels vs functional outcomes would be to make some correlation plots.

This is a great idea and now included as new Figure 9. Please see details in our response to your question 1.

We are pleased to inform you that your manuscript is accepted for publication and is now being sent to our publisher to be included in the next available issue of EMBO Molecular Medicine.

Corresponding Author Name: Debra Abramov and Jacqueline Burré

Manuscript Number: EMM-2020-12354